# *Leucine aminopeptidase1* controls egg deposition and hatchability in male *Aedes aegypti* mosquitoes

Xiaomei Sun [1,2,5], Xueli Wang [1,2,5], Kai Shi[1,2], Xiangyang Lyu[1,2], Jian Sun [3], Alexander S. Raikhel[4] & Zhen Zou [1,2] ✉

*Aedes aegypti* are vectors for several arboviruses infecting hundreds of millions of people annually. Controlling mosquito populations by regulating their reproduction is a potential strategy to minimize viral transmission in the absence of effective antiviral therapies or vaccines. Here, we demonstrate that leucine aminopeptidase1 (LAP1), detected by a SWATH-MS-based proteomic screen of female spermathecae, is a crucial determinant in mosquito population expansion. Mitochondrial defects and aberrant autophagy of sperm in *LAP1* mutant males (*LAP1$^{-/-}$*), prepared using CRISPR/Cas9 system, result in a reduction of reproduction in wild-type females that mated with them. The fitness of *LAP1$^{-/-}$* males is strong enough to efficiently transmit genetic changes to mosquito populations through a low number of hatchable offspring. Thus, *LAP1$^{-/-}$* males represent an opportunity to suppress mosquito populations and further studies should be undertaken to characterize *LAP1*'s suitability for gene drive usage.

*Aedes aegypti* is the primary transmitter of yellow fever virus (YFV), dengue virus (DENV), chikungunya virus (CHIKV), and Zika virus (ZIKV), and these infections cause the extreme suffering of hundreds of thousands of people[1]. Despite efforts to develop effective countermeasures against viral infections, antiviral therapies have not been approved for the treatment of infected individuals or to prevent infection in high-risk populations[2]. Current control of the *Aedes* vectors thus largely relies on the use of insecticides. Nevertheless, the rapid spread of insecticide resistance increases the difficulty of vector control worldwide[3]. Alternative strategies for mosquito population control are thus warranted.

Recently, *Wolbachia* infection, sterile insect techniques (SITs), and a genetic drive have emerged as promising approaches to suppress mosquito populations. *W. pipientis* is a maternally inherited intracellular bacterium that can be employed to induce conditional sterility between released infected males and wild-type (WT) females via cytoplasmic incompatibility (CI)[4,5]. While SIT depends on male

sterilization by radiation or chemicals, CRISPR/Cas9 systems, and release of insects carrying a dominant lethal (RIDL), are species-specific and safe methods of biological insect control that rely on the release of numerous sterile insects into the environment to suppress populations and minimize disease transmission[6]. In *Ae. aegypti*, it has developed a precision-guided sterile insect technique (pgSIT), which could generate flightless females and sterile males using a CRISPR-based approach[7]. Moreover, emerging genetic drive systems based on CRISPR/Cas9 could enforce inheritance in a hyper-Mendelian manner and rapidly spread target genes in the population[8]. Hence, a thorough investigation of mosquito reproduction would provide valuable cues for screening CRISPR/Cas9-target genes to reduce mosquito populations.

Many proteases have emerged as prominent reproductive players in insects and mammals[9,10]. Leucine aminopeptidases (LAPs), which are members of the metalloprotease family, are capable of cleaving leucine residues located at the N-terminal end of peptides or proteins. In

[1]State Key Laboratory of Integrated Management of Pest Insects and Rodents, Institute of Zoology, Chinese Academy of Sciences, Beijing 100101, China. [2]CAS Center for Excellence in Biotic Interactions, University of Chinese Academy of Sciences, Beijing 100049, China. [3]Key Laboratory of Zoological Systematics and Evolution, Institute of Zoology, Chinese Academy of Sciences, Beijing 100101, China. [4]Department of Entomology, University of California, Riverside, CA 92521, USA. [5]These authors contributed equally: Xiaomei Sun, Xueli Wang. ✉e-mail: zouzhen@ioz.ac.cn

addition to their key role in peptide turnover, LAPs also have various functions in bacteria, mammals, and plants[11–13]. Notably, the neofunctionalization of LAPs have been demonstrated in sperm. In mammals, LAP activity in seminal plasma correlated with male fertility[14]. Furthermore, proteomic analysis revealed a high abundance of LAP orthologs of sperm in Lepidoptera, *Drosophila*, and diverse mammalian taxa (human, rat, mouse, and *Macaca mulatta*)[15–20].

In *Drosophila*, eight LAP proteins (S-LAPs) were identified in the sperm proteome and have been proven to be essential to spermatogenesis. Additionally, *LAP* mutations result in male sterility by disrupting the accumulation of paracrystalline material and the structure of primary mitochondrial derivatives in *Drosophila* spermatids[21]. Cytosol aminopeptidases (AAEL006975, AAEL000108, and AAEL023987), the orthologs of the eight LAPs in *Drosophila*, were among the top 10 most abundant sperm proteins in *Ae. aegypti*[22]. Although the reproductive function of LAP in sperm is studied in mammals and *Drosophila*, little is known about this in other insect species, including mosquitoes.

Mosquito sperm consists of the head with nucleus and a functional flagellum, which is composed of two mitochondrial derivatives and an axoneme, a 9 + 9 + 1 microtubular structure responsible for sperm motility[23]. Mitochondria, the major organelles that produce cellular energy and reactive oxygen species (ROS), play an important role in regulating various physiological aspects of reproductive function, from spermatogenesis to fertilization. Functional defects in the mitochondria severely impair the production of energy that is required to maintain sperm motility[24].

A single mating enables a female mosquito to store enough sperm in the spermathecae to contribute to fertilizing all the eggs in the female lifetime. Males transfer a mixture of seminal fluid and sperm directly into female spermathecae through insemination; the sperm then moves along the spermathecal ducts and are then released to fertilize eggs during ovulation[25]. Structurally, there are three functional spermathecae in female *Ae. aegypti*: one large and two small[26]. Functionally, the secretions of spermathecal glands—containing energy-metabolism enzymes, glycoproteins, and lipoproteins—are responsible for sustaining and nourishing sperm that are stored in spermathecae[27,28]. Furthermore, spermathecae protect sperm from oxidative stress and damage contributing to sperm viability and long-term storage[27,29]. Previous studies have shown that female fecundity is specifically reduced by double-stranded RNA (dsRNA) knockdown of *glucose dehydrogenase* (*Gld*), *N-acetylgalactosaminyl transferase 6* (*GALNT6*), or *Kazal-type serine protease inhibitor* (*KSPI*), which were selected by RNA-seq analysis between virgin and inseminated female spermathecae[30]. Therefore, proteins related to fertility and fecundity in spermathecae may be potentially utilized as CRISPR/Cas9-targeted genes to control mosquito populations.

Although the structure and function of spermathecae have been well characterized, our current understanding of reproduction-related proteins in mosquito spermathecae is limited. In this study, we combined proteomic and bioinformatic analysis to identify that LAP1 is enriched in both testes and spermathecae. We then demonstrate that LAP1 deficiency disrupts the normal structure and development of sperm in male mosquitoes, which in turn suppresses the fertility and fecundity of females that mated with them. This study sheds light on the molecular complexity of LAP1 in relation to sperm function and provides a genetic strategy to target *LAP1* for mosquito population control.

## Results

### Analysis of spermathecal fluid proteome by SWATH-MS in *Ae. aegypti*

To identify reproduction-related proteins in the spermathecae of *Ae. aegypti*, the tissue was collected for the sequential window acquisition of all theoretical fragment ion spectra mass spectrometry (SWATH-MS) analysis, which is a data-independent acquisition (DIA) technology that can fully measure and quantify virtually all detectable compounds in a sample (Supplementary Fig. 1a). The female mosquitoes used in this analysis included unmated (virgin) females at 72 h post eclosion (PE), mated at 72 h PE and mated at 5 days post blood meal (PBM), designated as G1, G2 and G3, respectively. After data processing, a pairwise comparative analysis of proteomic data was conducted among G2/G1, G3/G1, and G3/G2. A total of 1,897 proteins were identified in these three groups. Volcano plots displayed the expression pattern of total proteins in three groups, consisting of upregulated (fold change > 1.5 and *P* < 0.05), downregulated (fold change < 0.667 and *P* < 0.05), and not significantly changed proteins. The protein levels of LAP1 (AAEL006975-PB) and DBF4-type zinc finger (DBF4, AAEL008779-PB) were shown to be significantly higher in the G2/G1 and G3/G1 groups; however, both were observed in the downregulated protein clusters of the G3/G2 (Fig. 1a and Supplementary Data 1). These findings suggest that these two proteins were derived because of mating. Subsequently, differentially expressed proteins (DEPs) in these three groups were selected for further analysis. The Venn diagram of DEPs revealed that 182 proteins were commonly regulated among the three groups, including 30, 25, and 28 upregulated proteins as well as 124, 73, and 83 downregulated proteins in G3/G1, G2/G1, and G3/G2 groups, respectively (Fig. 1b and Supplementary Data 2). Taken together, these results indicated that mating and oogenesis influence the protein levels in the spermathecae.

To prove the functional classification of the DEPs in these three groups, 182 co-existing proteins were used to perform Gene Ontology (GO) and the Kyoto Encyclopedia of Genes and Genomes (KEGG) analyses. Interestingly, proteins enriched with GO terms associated with cytoskeleton (GO:0005200), microtubule (GO:0005874) and microtubule-based process (GO:0007017) were all upregulated (Supplementary Fig. 1b). The KEGG results indicated that compared with other pathways, upregulated proteins were mainly enriched in the phagosome, as well as metabolic and glycolysis/gluconeogenesis pathways (Supplementary Fig. 1c). Next, we investigated how mating behavior regulates reproduction and therefore selected 25 upregulated proteins in the G2/G1 group for Sankey dot analysis. The results demonstrated that proteins were overrepresented in the functional categories of cytoplasm (Fig. 1c and Supplementary Table 1). Subsequently, LAP1, DBF4, and eight other candidate proteins were selected for further functional verification (Supplementary Table 1).

### CRISPR/Cas9-mediated deletion of *LAP1* affects the fecundity and fertility of female mosquitoes

To confirm the effect of ten selected proteins on population suppression, we knocked down these genes using RNA interference (RNAi) in virgin females, and these mosquitoes were then mated with males for 3 days (Supplementary Table 2). The results demonstrated that both fecundity (egg deposition) and fertility (hatchability) were suppressed in *M12 mutant protein precursor* (*M12*) and *LAP1* dsRNA-treated mosquitoes compared with *enhanced green fluorescent protein* (*EGFP*) dsRNA-treated mosquitoes (Fig. 2a). Silencing *sodium bicarbonate cotransporter* (*Sbc*) and *thioredoxin peroxidase* (*TPX5*) only decreased mosquito egg deposition, and silencing *chloride channel protein* (*Ccp*) only decreased mosquito hatchability (Fig. 2a). However, silencing *DBF4*, which was highest in the G2/G1 and G3/G1 groups, had no noticeable effect on the egg deposition and hatchability (Fig. 2a). Considering the results of bioinformatics analysis described above, we hypothesized that LAP1 plays a pivotal role in reproduction of female *Ae. aegypti*. To verify our hypothesis, we generated *LAP1* mutant (*LAP1*⁻/⁻) using the CRISPR/Cas9 approach. Sequencing using PCR showed that the protein-coding sequence was disrupted by the deletion of 11 bases in the third exon (Supplementary Fig. 2a). Furthermore, the LAP1 protein was not detected using western blot analysis of *LAP1*⁻/⁻ mutants (Supplementary Fig. 2b).

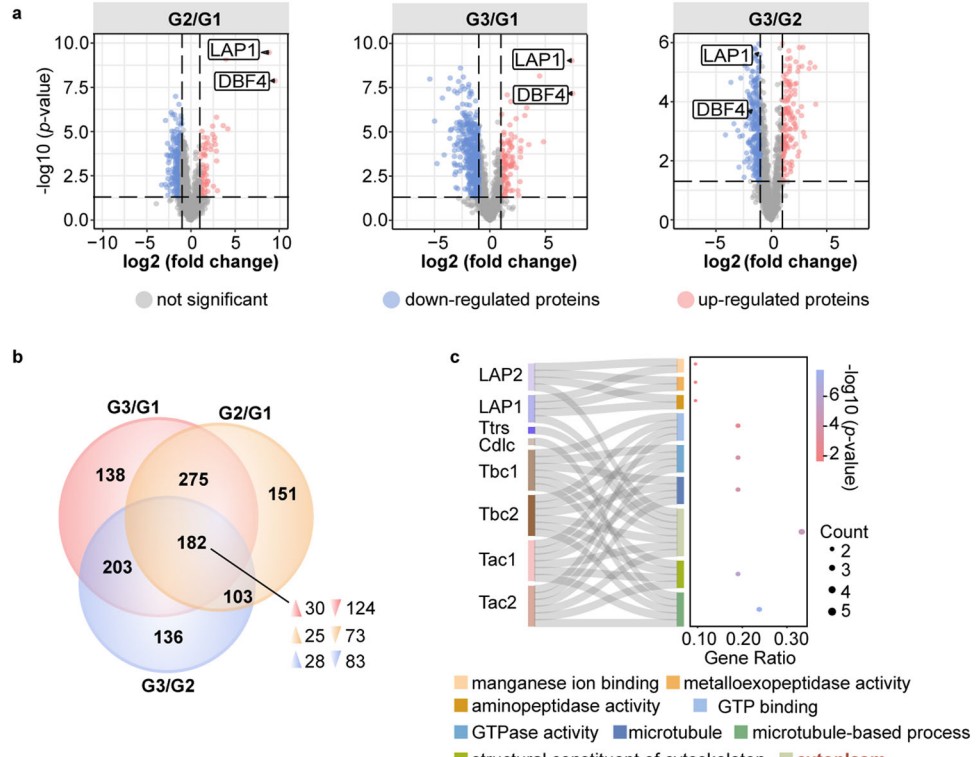

**Fig. 1 | Comparative analysis of the spermathecal fluid proteome in *Aedes aegypti*. a** Volcano plot analysis of total proteins in three groups. Up-regulated, down-regulated, and non-significantly regulated proteins are marked with red, blue and gray spots, respectively. Candidates (LAP1 and DBF4) discussed in the text are highlighted (filled black triangles). LAP1: leucine aminopeptidase1 (AAEL006975-PB), DBF4: DBF4-type zinc finger (AAEL008779-PB). Fold change is the ratio between the mean quantitation values of protein. Statistical significance was determined using a two-sided unpaired *t*-test. **b** Venn diagram showing the uniquely and commonly regulated differentially expressed proteins (DEPs). The direction of arrows in the overlapping region represents up- (fold change > 2 and $P < 0.05$) and down-regulated (fold change < 0.5 and $P < 0.05$) proteins. Statistical significance was determined using a two-sided unpaired *t*-test. **c** The Sankey dot plot of significantly up-regulated proteins in the G2/G1 group. Columns of Sankey represent proteins name (the left columns) and corresponding GO terms (the right columns). Terms are shown below the plot. The sizes of dot plot indicate counts of proteins in the corresponding term and the colors represent the *p*-value. The DAVID web server was used for GO enrichment analysis and the *p*-value was determined by a two-sided Fisher's exact test with Benjamini–Hochberg adjustment.

Then, we performed the mating assay to further determine the role of LAP1 in *Ae. aegypti* reproduction. Mosquitoes were separated into three groups (WT♂ × WT♀, *LAP1*⁻/⁻♂ × *LAP1*⁻/⁻♀, and *LAP1*⁻/⁺♂ × *LAP1*⁻/⁺♀) and mated for 3 days. Next, egg deposition and hatchability were assessed after the first and second blood meals, which were performed following the plan detailed in Supplementary Fig. 2c. The results revealed that knocking out *LAP1* resulted in a higher inhibition of mosquito hatchability (decreased by 90% and 100% after the first and second blood meals, respectively) compared to its effect on egg-laying (decreased by 28.5% and 38% after the first and second blood meals, respectively), suggesting that the effect of LAP1 on embryonic development may be greater than that on ovarian development in mosquitoes (Fig. 2b–e). Notably, the effect of LAP1 on mosquito reproduction in *LAP1*⁻/⁺ heterozygotes was weaker than that in *LAP1*⁻/⁻ homozygotes (Fig. 2c–e). In addition, 61% of the *LAP1*⁻/⁻♂ × *LAP1*⁻/⁻ ♀ group laid hatchable eggs after the first blood meal, while none of laid eggs hatched after the second (Fig. 2f, g). This suggests that the effect of *LAP1* deletion on reproduction became stronger over time.

To determine whether the role of LAP1 depends on *LAP1*⁻/⁻ males or *LAP1*⁻/⁻ females, mosquitoes were separated into three groups (WT♂ × WT♀, *LAP1*⁻/⁻♂ × WT♀, and WT♂×*LAP1*⁻/⁻♀). Similarly, egg deposition and hatchability were measured after the first and second blood meals (Supplementary Fig. 2d). As shown in Fig. 2h, j, female fecundity and fertility markedly decreased in both *LAP1*⁻/⁻♂ × WT♀ and WT♂×*LAP1*⁻/⁻♀ groups after the first blood meal. After the second

blood meals, although the egg deposition of WT♂×*LAP1*⁻/⁻♀ group was notably lower than in WT♂ × WT♀, there were no obvious changes in the *LAP1*⁻/⁻♂ × WT♀ group (Fig. 2i). However, hatchability in *LAP1*⁻/⁻♂ × WT♀ (2.9%) was significantly lower than that in WT♂×*LAP1*⁻/⁻♀ (55.74%) (Fig. 2k). Furthermore, the degree of decline in hatching rate in the *LAP1*⁻/⁻♂ × WT♀ group was much greater than that in the WT♂ × *LAP1*⁻/⁻♀ group, not only after the first blood meal but also after the second (Fig. 2j, k). Consistently, the percentage of hatchable eggs laid by mosquitoes decreased from 69% after the first blood meal to 50% after the second blood meal in *LAP1*⁻/⁻♂ × WT♀, whereas it remained unchanged in the WT♂×*LAP1*⁻/⁻♀ group (Fig. 2l, m). These findings suggest that the CRISPR/Cas9-mediated knockout of *LAP1* had a profound effect on reproduction in males relative to females, and LAP1 might primarily have an impact on the function of sperm rather than egg. Taken together, these results strongly suggest that the presence of LAP1 is essential for reproduction and sperm function in *Ae. aegypti*.

### LAP1 proteins localize to *Ae. aegypti* sperm

To further determine the function of LAP1 on reproduction, we investigated the expression profiles of *LAP1* using quantitative PCR (qPCR), and the results indicated it is mainly expressed in testis (Fig. 3a). Subsequently, an immunofluorescence assay was performed to localize LAP1 in sperm of WT, *LAP1*⁻/⁺, and *LAP1*⁻/⁻ male mosquitoes. We observed that LAP1 was distributed in the head and tail of sperm in the WT males. The density of the LAP1 immunofluorescent signal,

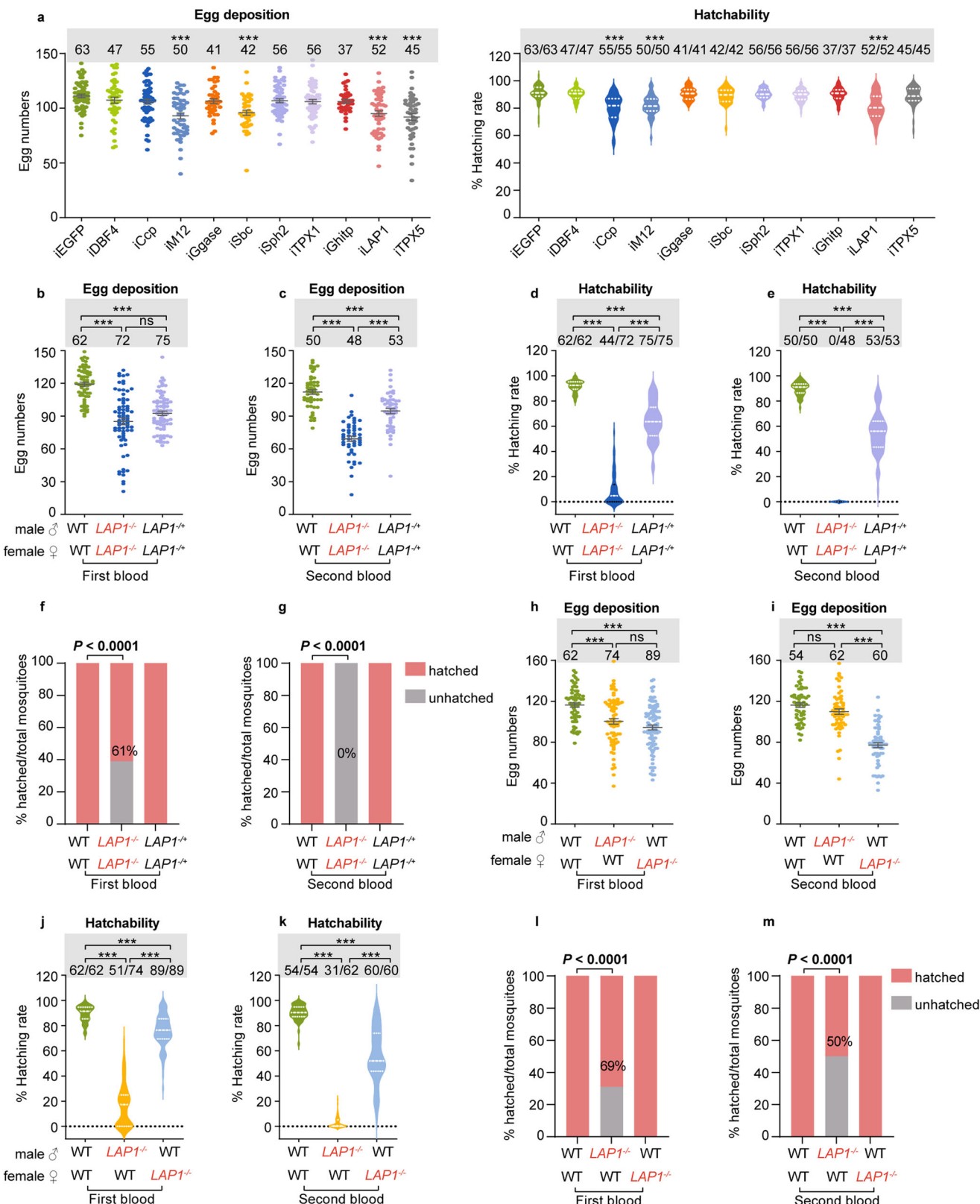

however, was notably lower in $LAP1^{-/+}$ male mosquitoes than WT, in which it was only present in the sperm head. Furthermore, the LAP1 immunofluorescent signal was barely detectable in sperm of $LAP1^{-/-}$ males (Fig. 3b). Meanwhile, we also found that LAP1 was absent in the spermathecae of virgin mosquitoes but highly expressed in the spermathecae of mated mosquitoes in both G2 and G3 (Supplementary Fig. 3). Overall, these results verify that LAP1 is principally located in the sperm of males, which in turn is transferred to the spermathecae of females with mating.

Because the above results showed that egg deposition and hatchability also decreased in the pair $WT\male \times LAP1^{-/-}\female$, we investigated whether the presence of LAP1 is essential for ovarian and egg development. To do so, the ovaries of females belonging to the $WT\male \times LAP1^{-/-}\female$ and $WT\female \times WT\male$ groups at 48 h PBM were dissected to

**Fig. 2 | _LAP1_ mutation in males limits female fecundity and fertility. a** Analysis of female fecundity and fertility after RNAi knockdown of ten candidate genes. dsRNA was microinjected into the thorax of 1-day-old virgin female mosquitoes (50 females per group). After a 3-day recovery period, females were mated with 50 males for 3 days before the blood meal. The numbers of eggs were counted at 6 days PBM, and subsequently these eggs were used to calculate the hatchability. Mosquitoes injected with _EGFP_ dsRNA were used as controls. The _q_-value was calculated by comparing the experimental group with the _EGFP_ dsRNA-treated group. _n_ = 63 for iEGFP; _n_ = 47 for iDBF4; _n_ = 55 for iCcp; _n_ = 50 for iM12; _n_ = 41 for iGgase; _n_ = 42 for iSbc; _n_ = 56 for both iSph2 and iTPX1; _n_ = 37 for iGhitp; _n_ = 52 for iLAP1; _n_ = 45 for iTPX5. **b, c** Comparison of the egg deposition among WT♂ × WT♀, _LAP1⁻ᐟ⁻_♂ × _LAP1⁻ᐟ⁻_♀, and _LAP1⁻ᐟ⁺_♂ × _LAP1⁻ᐟ⁺_♀ groups after the first (**b**; _n_ = 62 for WT♂ × WT♀; _n_ = 72 for _LAP1⁻ᐟ⁻_♂ × _LAP1⁻ᐟ⁻_♀; _n_ = 75 for _LAP1⁻ᐟ⁺_♂ × _LAP1⁻ᐟ⁺_♀) and second (**c**; _n_ = 50 for WT♂ × WT♀; _n_ = 48 for _LAP1⁻ᐟ⁻_♂ × _LAP1⁻ᐟ⁻_♀; _n_ = 53 for _LAP1⁻ᐟ⁺_♂ × _LAP1⁻ᐟ⁺_♀) blood meals (50 females were paired with 50 males per group). **d–g** Comparison of the hatchability among WT♂ × WT♀, _LAP1⁻ᐟ⁻_♂ × _LAP1⁻ᐟ⁻_♀, and _LAP1⁻ᐟ⁺_♂ × _LAP1⁻ᐟ⁺_♀ groups after the first (**d** and **f**) and second (**e** and **g**) blood meals (50 females were paired with 50 males per group). For **d**, _n_ = 62 for WT♂ × WT♀; _n_ = 72 for _LAP1⁻ᐟ⁻_♂ × _LAP1⁻ᐟ⁻_♀; _n_ = 75 for _LAP1⁻ᐟ⁺_♂ × _LAP1⁻ᐟ⁺_♀. For **e**, _n_ = 50 for WT♂ × WT♀; _n_ = 48 for _LAP1⁻ᐟ⁻_♂ × _LAP1⁻ᐟ⁻_♀; _n_ = 53 for _LAP1⁻ᐟ⁺_♂ × _LAP1⁻ᐟ⁺_♀. The percentages in **f** and **g** indicate the rate that mosquitoes produce hatchable eggs. **h, i** Comparison of the fecundity among WT♂ × WT♀, _LAP1⁻ᐟ⁻_♂ × WT♀, and WT♂ × _LAP1⁻ᐟ⁻_♀ groups after the first (**h**; _n_ = 62 for WT♂ × WT♀; _n_ = 74 for _LAP1⁻ᐟ⁻_♂ × WT♀; _n_ = 89 for WT♂ × _LAP1⁻ᐟ⁻_♀) and second (**i**; _n_ = 54 for WT♂ × WT♀; _n_ = 62 for _LAP1⁻ᐟ⁻_♂ × WT♀; _n_ = 60 for WT♂ × _LAP1⁻ᐟ⁻_♀) blood meals (50 females were paired with 50 males per group). **j–m** Comparison of the fertility among WT♂ × WT♀, _LAP1⁻ᐟ⁻_♂ × WT♀, and WT♂ × _LAP1⁻ᐟ⁻_♀ groups after the first (**j** and **l**) and second (**k** and **m**) blood meals (50 females were paired with 50 males per group). For **j**, _n_ = 62 for WT♂ × WT♀; _n_ = 74 for _LAP1⁻ᐟ⁻_♂ × WT♀; _n_ = 89 for WT♂ × _LAP1⁻ᐟ⁻_♀. For **k**, _n_ = 54 for WT♂ × WT♀; _n_ = 62 for _LAP1⁻ᐟ⁻_♂ × WT♀; _n_ = 60 for WT♂ × _LAP1⁻ᐟ⁻_♀. The percentages in **l** and **m** indicate the rate that mosquitoes produce hatchable eggs. The top of each column in **b**, **c**, **h**, and **i** indicates the numbers of mosquitoes that laid eggs, while in **d**, **e**, **j** and **k** shows the numbers of hatched mosquitoes relative to total egg-laying mosquitoes. Females that died during egg laying were removed from the study. Each dot represents one mosquito. The experiments were independently performed twice with 50 females per experiment. A two-sided Mann−Whitney test of variance was conducted to detect any significant variation between two replicates, and no significant differences were detected. Then results were combined for further analyses. Statistical significance was analyzed using the Kruskal−Wallis test followed by Dunn's post-hoc tests with Benjamini−Hochberg adjustment for multiple comparisons (**a−e** and **h−k**) and two-sided Fisher's exact test (**f**, **g**, **l**, and **m**). Data in **a−e** and **h−k** are shown as mean ± SEM. *_q_ < 0.05, **_q_ < 0.01, ***_q_ < 0.001. ns not significant. Source data are provided as a Source Data file.

assess ovarian development. As indicated in Fig. 3c, the ovaries of the _LAP1⁻ᐟ⁻_ females failed to undergo normal development, and the length of the ovarian follicles was 29.9% (295.8 μm on average) lower than in WT mosquitoes (422 μm on average). In addition, the WT mosquitoes almost completed digestion at 48 h PBM, whereas the midguts of _LAP1⁻ᐟ⁻_ mosquitoes were still full with blood, which might account for low egg deposition (Fig. 3c). However, the detailed molecular mechanism underlying this phenotype requires further exploration. Subsequently, eggs from the pairs _LAP1⁻ᐟ⁻_♂ × WT♀ and WT♂ × WT♀ were collected at different periods (0−1, 1−2, 2−4, 4−6, 6−8, 8−10, 10−12, and 20−22 h) after the first and second blood meals. Afterwards, RNA was extracted for qPCR and the results did not reveal any significant differences in _LAP1_ expression between the two groups (Fig. 3d). These findings suggest that LAP1 exerted its effect on egg development and suppressed female fertility by regulating sperm development.

## Mutation of _LAP1_ results in mitochondrial derivative deficiency and activation of autophagy in the testes

Mature sperm in most insects consists of an elongated cell with an apical acrosome inserted into the pronuclear region, a long nucleus occupying the anterior spermatozoa, and a posterior long flagellum (Supplementary Fig. 4a)[31]. Mitochondrial derivatives with or without dense fibers filled most of the volume of the mature sperm in insects, which includes a two-part spherical mass that elongates along the entire length of the tail and remains closely associated with the axoneme (Supplementary Fig. 4a)[32,33]. In this study, we investigated whether the LAP1 protein plays an important role in sperm mitochondrial development. Sperm at 48 h PE from WT and _LAP1⁻ᐟ⁻_ testes were collected for immunofluorescence assay. The results showed that LAP1 proteins co-localized with the mitochondria in sperm of WT males. However, the immunofluorescence signals of both LAP1 protein and mitochondria were almost undetectable in _LAP1⁻ᐟ⁻_ sperm, which suggest that LAP1 is associated with mitochondrial development (Fig. 4a).

To determine how the LAP1 deficiency hindered sperm function, transmission electron microscopic (TEM) analysis was performed to examine ultrastructural changes (axoneme, mitochondria, and microtubules) of sperm in WT and _LAP1⁻ᐟ⁻_ mosquito testes. The transverse sections of sperm showed that most sperm had intact cell membranes, and the elliptical mitochondrial derivatives were approximately equal in size and regularly arranged in WT testes (Fig. 4b: WT_A-WT_G and Supplementary Fig. 4b). Conversely, the elliptical mitochondrial derivatives in _LAP1⁻ᐟ⁻_ testes were markedly atrophied, and the sperm cell membrane ruptured, resulting in the extracellular dissociation of the 9 + 9 + 1 microtubule structures (Fig. 4b: _LAP1⁻ᐟ⁻_ _A-_LAP1⁻ᐟ⁻_ _G and Supplementary Fig. 4b). Importantly, we detected the autophagic process in _LAP1⁻ᐟ⁻_ testes. First, the elliptical mitochondrial derivatives of _LAP1⁻ᐟ⁻_ mosquitoes were markedly atrophied and surrounded by double membrane-limited autophagosome vesicles (Fig. 4c: _LAP1⁻ᐟ⁻_ _A, _LAP1⁻ᐟ⁻_ _A1, _LAP1⁻ᐟ⁻_ _B, and _LAP1⁻ᐟ⁻_ _B1). The 9 + 9 + 1 structures disintegrated or were released extracellularly (Fig. 4c: _LAP1⁻ᐟ⁻_ _A2 and _LAP1⁻ᐟ⁻_ _A3). Finally, the degraded vesicular material and mitochondrial remnants around disintegrated sperm structures were engulfed by autophagosomes (Fig. 4c: _LAP1⁻ᐟ⁻_ _B2-_LAP1⁻ᐟ⁻_ _B3 and Supplementary Fig. 4c). In contrast, no autophagosomes were observed in WT testes. Collectively, TEM data revealed that mutation of LAP1 induced the production of defective mitochondrial derivatives and the activation of autophagy in sperm.

### _LAP1_ mutation leads to the reduced sperm fertility

Due to the difference in the egg hatchability in fertile females from the _LAP1⁻ᐟ⁻_♂ × WT♀ group between the first and second blood meals (Fig. 2j, k), we aimed to understand whether blood feeding for the indicated days after mating influenced hatching rate. Mosquitoes were mated for 3 days in the _LAP1⁻ᐟ⁻_♂ × WT♀ and WT♂ × WT♀ groups, and then males were removed, and the females were allowed to have a blood feeding at various days (0, 4, 8, 12, and 16 days) post mating to measure hatchability (Fig. 5a). We found that delaying for 4 days suppressed the mean mosquito hatching rate from 13.7% to 6.1% in the _LAP1⁻ᐟ⁻_♂ × WT♀ group, a much lower rate than the 90% observed for the WT♂ × WT♀ group (Fig. 5b). Delaying to 8 days and 12 days led to an even greater suppression (2.4% and 2.1%, respectively); and in females that were fed with blood at 16 days after mating, fertility was almost eliminated in the experimental group (0.88%) (Fig. 5b). Fried's Competitiveness Index was used to quantify the mating competitiveness of _LAP1⁻ᐟ⁻_ males in blood-feeding experiments on females for indicated days (0, 4, 8, 12, and 16 days after mating), and it was 0.91, 0.97, 1.05, 1.05, and 1.09, respectively, indicating that the mating competitiveness of _LAP1⁻ᐟ⁻_♂ and WT♂ was comparable. Furthermore, curve fitting showed that females were allowed to have a blood feeding at 0.88 days post mating achieved 10% hatchability (Fig. 5c). These results indicated that female fertility was limited by number of days after mating, and fertility capacity of stored sperm in females decreased over time.

Subsequently, to verify whether the fertilizing ability of sperm in _LAP1⁻ᐟ⁻_ males changes over time, another mated assay was performed. Briefly, WT females were mated with indicated day-old WT and _LAP1⁻ᐟ⁻_

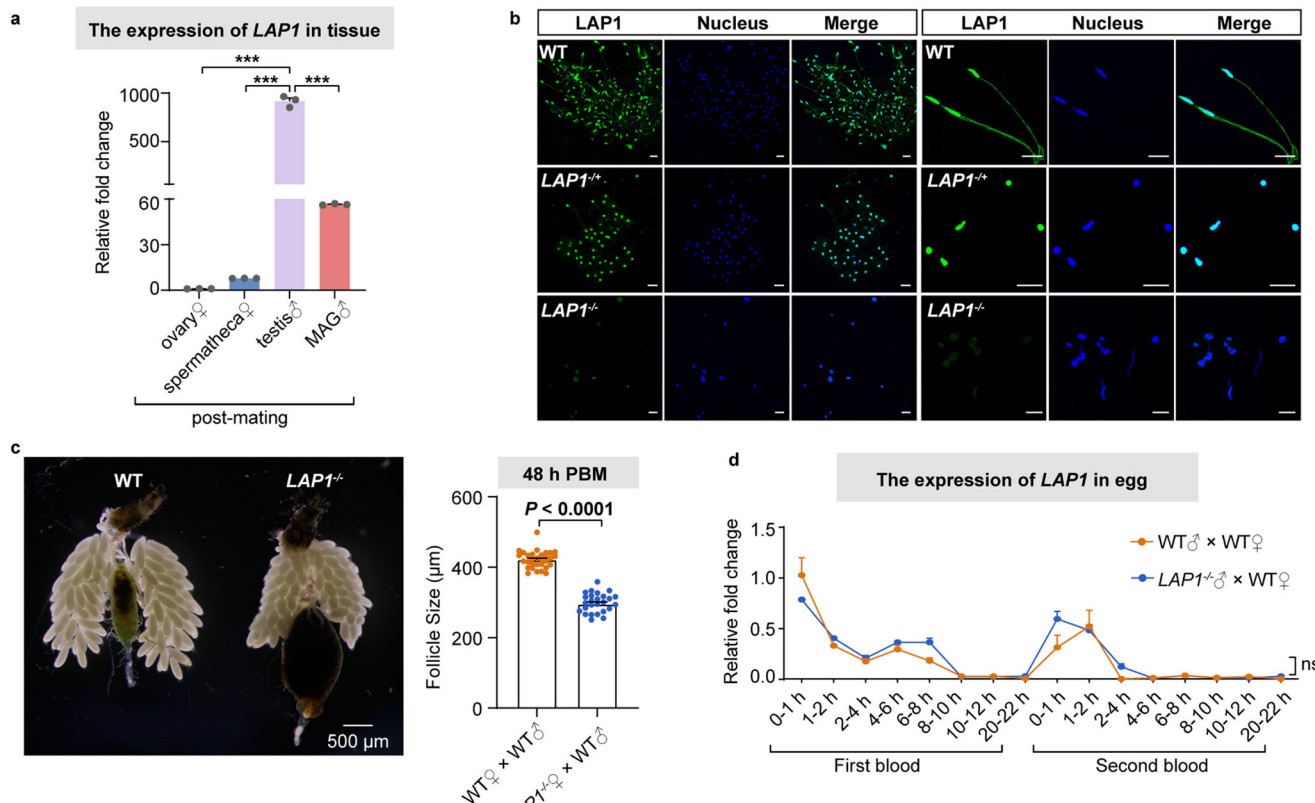

**Fig. 3 | The localization of LAP1 in *Ae. aegypti*. a** mRNA abundance of *LAP1* in different mosquito tissue at the G2 (*n* = 3). The experiment was replicated three times, and tissue collected from 20 individual females and males at 72 h PE were served as a single replicate per group. Data were normalized to the expression level of ovary♀. Statistical significance was determined using the one-way ANOVA test followed by Tukey's post hoc tests with Benjamini−Hochberg adjustment for multiple comparisons. \*\*\**q* < 0.001. **b** Images of sperm in WT, *LAP1⁻/⁺*, and *LAP1⁻/⁻* male mosquitoes. Sperm was dissected, and LAP1 protein was detected by immunofluorescence assay using mouse anti-LAP1 antibody (green). Nucleus was stained by Hoechst 33258 (blue). Images were visualized under a confocal microscope. Scale bar: 20 μm. **c** The ovarian phenotypes (left panel) and follicle lengths (right panel) of WT and *LAP1⁻/⁻* female mosquitoes were detected at 48 h PBM (*n* = 31 for WT♀ × WT♂; *n* = 26 for *LAP1⁻/⁻*♀ × WT♂). Results are combined from three batches of mosquitoes, and each dot represents the length of a single follicle. Images were captured from CellSens software (version 1.6) using an Olympus

SZX16 stereoscopic microscope at 5 × magnification (scale bar: 500 μm). The follicle lengths were measured in CellSens software (version 1.6). The *p*-value was determined by the two-sided Mann−Whitney test. **d** Temporal expression of *LAP1* during different stages of embryonic development in *LAP1⁻/⁻*♂ × WT♀ and WT♂ × WT♀ groups (*n* = 3). The experiment was repeated three times, and total RNA was extracted from eggs collected from the pairs *LAP1⁻/⁻*♂ × WT♀ and WT♂ × WT♀ at different periods (0–1, 1–2, 2–4, 4–6, 6–8, 8–10, 10–12, and 20–22 h) after the first and second blood meals as individual replicates for each group. RNA level of *LAP1* was measured using qPCR. Data were normalized to the expression level of WT♂ × WT♀ at 0–1 h post first blood meal. Statistical significance was determined using the two-sided multiple Mann−Whitney test with Benjamini−Hochberg adjustment. ns: not significant. Data in **a**, **c** and **d** are represented as mean ± SEM. The experiments were repeated three times with similar results. Source data are provided as a Source Data file.

males (1-, 3-, 5-, 7-, 9-, and 11-day-old males). And blood meal was given after mating for 3 days. Egg deposition and hatchability were measured later (Fig. 5d). Females that were mated with 1- and 3-day-old *LAP1⁻/⁻* males demonstrated a decreased mean hatching rate from around 90% to 17.7% and 16.6%, respectively, while mating with 5- and 7-day-old *LAP1⁻/⁻* males resulted in a much greater inhibition (to 11.4% and 9.1%) (Fig. 5e). Importantly, mating with 9- and 11-day-old *LAP1⁻/⁻* males either almost abolished female fertility or strongly suppressed oviposition (5.9% and 4.6%) (Fig. 5e). We also measured the mating competitiveness of *LAP1⁻/⁻* males in mating experiments between females and indicated age mutants using Fried's Competitiveness Index, which was 1.05, 1.77, 1.90, 1.90, 1.69, and 1.85 for mating with 1-, 3-, 5-, 7-, 9-, and 11-day-old males, respectively, indicating that *LAP1⁻/⁻*♂ were comparable to or more competitive than WT♂ for mating. In addition, curve fitting demonstrated that mating with an average of 5.71-day-old *LAP1⁻/⁻* males resulted in a hatchability that was only 10% of that in the control group (Fig. 5f). Taken together, these results reveal that the effect of LAP1 on the fertile capacity of sperm produced by males or stored by females became stronger over time, suggesting that we

would be able to modulate hatchability by controlling the blood-feeding time of females that were mated with *LAP1⁻/⁻* males or the time when females mated with *LAP1⁻/⁻* males.

## Fitness and safety assessments of *LAP1⁻/⁻* males

SIT involves inundating a local insect population with sterile males, thereby suppressing female fertility[7]. Therefore, the major requirement for the successful application of SIT is the ability to provide large numbers of active and safe genetically sterile males to wild populations on a large scale[34]. To assess the ability of *LAP1⁻/⁻* males to compete with WT males for mating, we introduced 30 virgin WT females along with newly emerged 1-day-old *LAP1⁻/⁻* and WT males, which were mixed at various ratios. The ratios of *LAP1⁻/⁻*♂ to WT ♂ were 0:1, 1:1, 2:1, 5:1, 9:1, 14:1, and 29:1 for a total of 30 males in each group (Fig. 6a). Blood meal was given after 3 days. And egg hatchability was measured. Adding a 1:1, 2:1, and 5:1 ratio resulted in a lower hatchability (68.5%, 66.2%, and 60.3%, respectively; hatchability at 0:1 ratio was 91.4% on average), while a 9:1 and 14:1 ratio showed a reduced hatchability to 46.6% and 38.5%, respectively. Importantly, the addition of 29:1 resulted in a

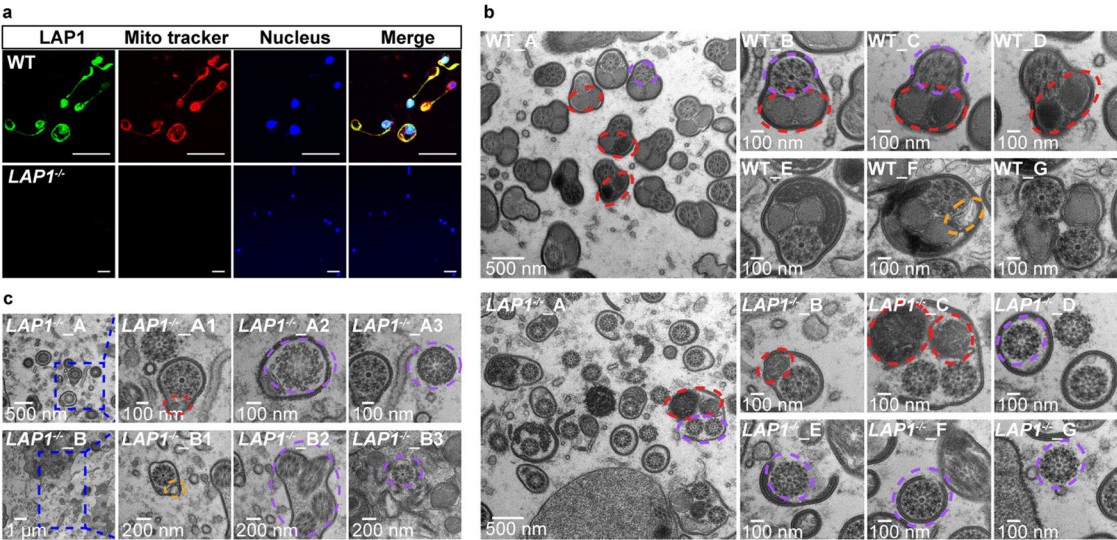

**Fig. 4 | CRISPR/Cas9-mediated deletion of *LAP1* impaired mitochondrial derivatives and activated autophagy in testes. a** Co-localization of LAP1 protein with mitochondria using immunofluorescence assay. LAP1 protein, nucleus, and mitochondria were stained by mouse anti-LAP1 antibody (green), Hoechst 33258 (blue), and Mito Tracker (red), respectively. Scale bar: 20 μm. **b** TEM analysis of the spermatozoa in WT (WT_A-WT_G) and *LAP1*⁻/⁻ (*LAP1*⁻/⁻_A-*LAP1*⁻/⁻_G) mosquitoes. The transverse sections of sperm contain two elliptical mitochondrial derivatives (red dashed line) and a 9 + 9 + 1 microtubule structure (purple dashed line) in testes. **c** TEM analysis revealed the autophagy process of mitochondria and microtubule structure in the *LAP1*⁻/⁻ spermatozoa. Compared with the WT mosquitoes, the elliptical mitochondrial derivatives of *LAP1*⁻/⁻ mosquitoes were distinctly irregular and atrophied, and autophagosome vesicles were formed close to the mitochondria as well as the structure of 9 + 9 + 1 tended to be disintegrated (*LAP1*⁻/⁻_A, *LAP1*⁻/⁻_A1-A3). Double-membrane-limited autophagosomes (blue dotted box) were filled with vesiculas and 9 + 9 + 1 structure of sperm (*LAP1*⁻/⁻_B and *LAP1*⁻/⁻_B1–B3). The scale bars were marked on the figure. Red ellipse dashed line: mitochondria. Purple ellipse dashed line: 9 + 9 + 1 structure. Yellow ellipse dashed line: autophagic vesicles. The experiments were repeated three times with similar results. Source data are provided as a Source Data file.

notable reduction in female hatchability (29:1 ratio, reduced to 33% on average) (Fig. 6b). In addition, curve fitting showed that an average of 8.34 1-day-old *LAP1*⁻/⁻ males were required to compete with one WT male to reduce female fertility by 50% (Fig. 6c).

Then, to further ensure the ability of *LAP1*⁻/⁻ males to suppress female fertility, 6-day-old *LAP1*⁻/⁻ males were selected for a competition assay using the same method as described above (Fig. 6d). The addition of a 1:1 ratio led to a significant decrease in female hatchability (0:1 ratio, 89.5% on average; 1:1 ratio, 64.1% on average), while a 2:1 ratio reduced hatchability to 52.8%, and a further ratio of 5:1 resulted in only a slight decrease in hatchability to 49.1%. Adding a 9:1 and 14:1 ratio suppressed the hatchability to 43.2% and 30.6%, respectively. Interestingly, the addition of 29:1 resulted in the lowest hatching rate in WT females (22.9% on average) (Fig. 6e). Moreover, curve fitting illustrated that the competition of 4.89 6-day-old *LAP1*⁻/⁻ males with one 1-day-old WT male resulted in a 50% reduction in WT female fertility (Fig. 6f).

Meanwhile, we also assessed physiological indicator and the safety of *LAP1*⁻/⁻ males under laboratory conditions—e.g., by assessing longevity and virus transmission ability—to improve the possibility and potential success of releasing *LAP1*⁻/⁻ males into wild mosquito populations. Fortunately, *LAP1* mutation produced no significant effect on the lifespan of male mosquitoes ($p = 0.22$) (Supplementary Fig. 5a). This facilitates the maintenance of mutant strains in the wild by *LAP1*⁻/⁻ males. Importantly, *LAP1* deletion did not alter the ability of *Ae. aegypti* to transmit ZIKV or DENV in comparison with WT mosquitoes, suggesting the relative biosafety of the *LAP1*⁻/⁻ mutants (Supplementary Fig. 5b). Based on all parameters measured above, *LAP1*⁻/⁻ males may be as safe and robust as WT males under laboratory conditions. Thus, it will be a promising strategy in the future to control female populations by mating with *LAP1*⁻/⁻ males of a given age.

## Discussion

Mosquito population control is considered as one of the most effective approaches to reduce the spread of mosquito-borne diseases.

Accordingly, female mosquitoes have been studied extensively due to the direct linkage between their intrinsic properties of blood feeding with egg development and disease pathogen transmission[35–43]. Male mosquitoes are essential for the expansion of mosquito populations as these provide fertile sperm during mating, which are then stored and maintained in the spermathecae[23]. Despite their importance, less attention has been paid to mosquito male biology.

Previous studies have confirmed that spermathecae, which promote sperm maturation and capacitation, have an extremely strong correlation with female fertility and fecundity[44]. Following copulation, sperm along with a mixture of seminal fluid move into spermathecae, where they are subsequently maintained and stored[25]. Thus, the composition and physiological conditions of spermathecae undergo dynamic changes to ensure sperm survival. Our proteomics-based analysis of mosquito spermathecae proteins has implied that the presence and content of sperm influence the expression of many proteins in the spermathecae (Fig. 1a). This result suggests that components and developmental environment in spermathecae undergo constant adjustment accompanied by the transfer and consumption of seminal fluid. Indeed, a previous study in honeybees elucidated a similar mechanism[29]. Furthermore, functional annotation of total protein cohorts indicated enrichment of categories related to cytoskeleton and microtubule synthesis in GO analysis, as well as identified phagosome, metabolic, and glycolysis/gluconeogenesis pathways by KEGG analysis (Supplementary Fig. 1b, c). Microtubules play indispensable roles in Sertoli and germ cell development, which may be the reason for the upregulation of proteins enriched with these GO terms after insemination[45,46]. Moreover, the mating-induced metabolic and glycolysis/gluconeogenesis regulation reflect that females provide ATP for sperm motility and survival during storage, which is consistent with previous studies in mice, *Drosophila*, and honeybees[47–49].

LAPs belong to the conserved M1 or M17 peptidase families and are distributed in all living organisms with wide specificity[13]. These have been reported to be the most abundant sperm proteins in *Ae.*

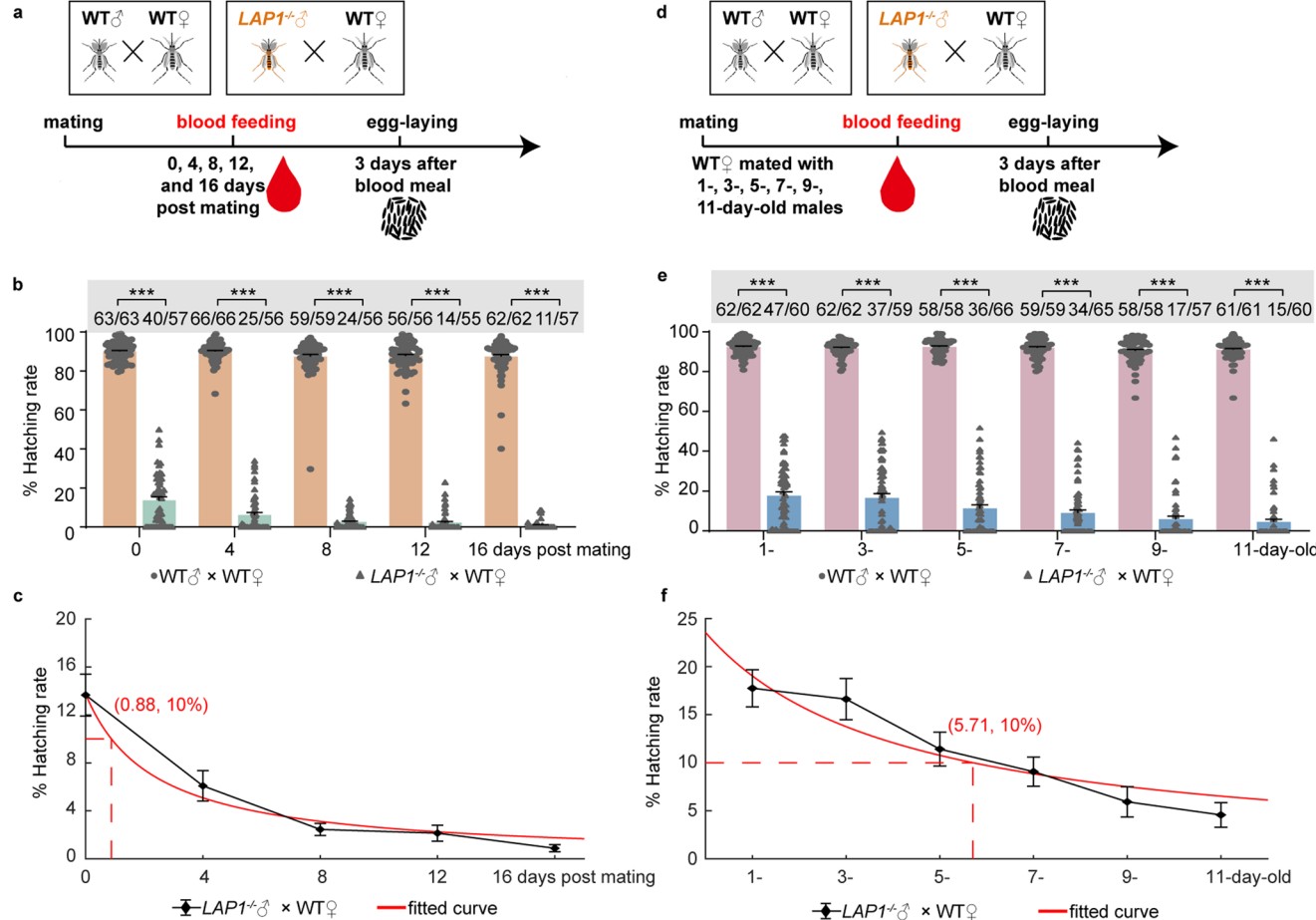

**Fig. 5 | Mating with *LAP1*−/− males suppresses female fertility. a** Schematic diagram of the stored sperm fertility assay. After mating with 1-day-old WT or *LAP1*−/− males for 3 days, WT females were blood fed at indicated days (0, 4, 8, 12, and 16 days) post mating. And egg hatchability was measured. **b** The fertility of WT females following blood feeding at indicated days after mating with 1-day-old WT or *LAP1*−/− males (30 females were paired with 30 males per group). For WT♂ × WT♀, *n* = 63 for 0 days post mating; *n* = 66 for 4 days post mating; *n* = 59 for 8 days post mating; *n* = 56 for 12 days post mating; *n* = 62 for 16 days post mating. For *LAP1*−/−♂ × WT♀, *n* = 57 for 0 days post mating; *n* = 56 for both 4 days post mating and 8 days post mating; *n* = 55 for 12 days post mating; *n* = 57 for 16 days post mating. The *q*-value was calculated by comparing the experimental group with the 0 days post mating group. **c** Curve fitting of the same data from **b**. The dashed line shows the point, at which mosquitoes were blood fed after mating with *LAP1*−/− males resulted in a 10% of female fertility. The red curve was served as the fitted curve. **d** Schematic diagram of the sperm fertility assay. WT females were mated with indicated day-old (1-, 3-, 5-, 7-, 9-, 11-day-old) WT or *LAP1*−/− males. And blood meal was given after mating for 3 days. Egg deposition and hatchability were measured later. **e** The fertility of WT females mated with indicated old-WT or *LAP1*−/− males (30 females were paired with 30 males per group). For WT♂ × WT♀, *n* = 62 for both 1-day-old

and 3-day-old; *n* = 58 for both 5-day-old and 9-day-old; *n* = 59 for 7-day-old; *n* = 61 for 11-day-old. For *LAP1*−/−♂ × WT♀, *n* = 60 for 1-day-old; *n* = 59 for 3-day-old; *n* = 66 for 5-day-old; *n* = 65 for 7-day-old; *n* = 57 for 9-day-old; *n* = 60 for 11-day-old. The *q*-value was calculated by comparing the experimental group with the 1-day-old group. **f** Curve fitting of the same data from **e**. The dashed line shows the point at which mating with indicated old-*LAP1*−/− males resulted in a 10% reduction in female fertility. The red curve was represented the fitted curve. Females that died during laying eggs were removed from the study. Three days after oviposition, the numbers of eggs were counted and subsequently separated into intact (unhatched) or broken (hatched). The top of each column in **b** and **e** shows the number of hatched mosquitoes relative to total mosquitoes. Each dot represents one mosquito. The experiments were independently performed three times with 30 females per experiment. A Kruskal–Wallis test of variance was conducted to detect any significant variation among three replicates, and no significant differences detected. Then results were combined for further analyses. Statistical significance was determined using the multiple Mann–Whitney test with Benjamini–Hochberg adjustment (**b** and **e**). Data are shown as mean ± SEM. *q* < 0.05, **q* < 0.01, ****q* < 0.001. Source data are provided as a Source Data file.

*aegypti*[22]. Concordant with previous studies, we also found that LAPs were significantly upregulated after insemination and enriched in GO terms—manganese ion binding, metalloexopeptidase activity, cytoplasm, and aminopeptidase activity—suggesting that LAPs serve a critical role in peptide turnover (Fig. 1c). In *Drosophila*, the LAP family was expanded and constituted the primary protein components of sperm[18]. Although LAP may not exert peptide turnover role in spermatogenesis due to loss of enzymatic activity during evolution, LAP is essential for the normal development of sperm[21]. In this study, we demonstrated that *LAP1* deficiency results in lower egg deposition, and this might occur due to poor blood digestion in the midgut (Figs. 2b and 3c)[50]. Importantly, *LAP1* male mutants suppressed sperm fertility.

When females mated with them, hatchability was markedly reduced after both first and second blood meals (Fig. 2d, e). Consistently, the effect of LAP1 in male fertility was similar to that reported in *Drosophila*[21]. Furthermore, this suggest that the abnormal seminal fluid caused by *LAP1* deletion from a single mating was sufficient to reduce female sterility.

Our studies showed that LAP1 localized to the sperm mitochondria, and its knockout resulted in defective mitochondrial derivatives (Fig. 3b). This might be responsible for the observed low hatchability because the rate of egg fertilization was low. Likewise, in *Drosophila* and silkworm, defects in mitochondrial derivatives led to male sterility by modulating sperm motility[21,51]. In humans, decreased sperm motility in

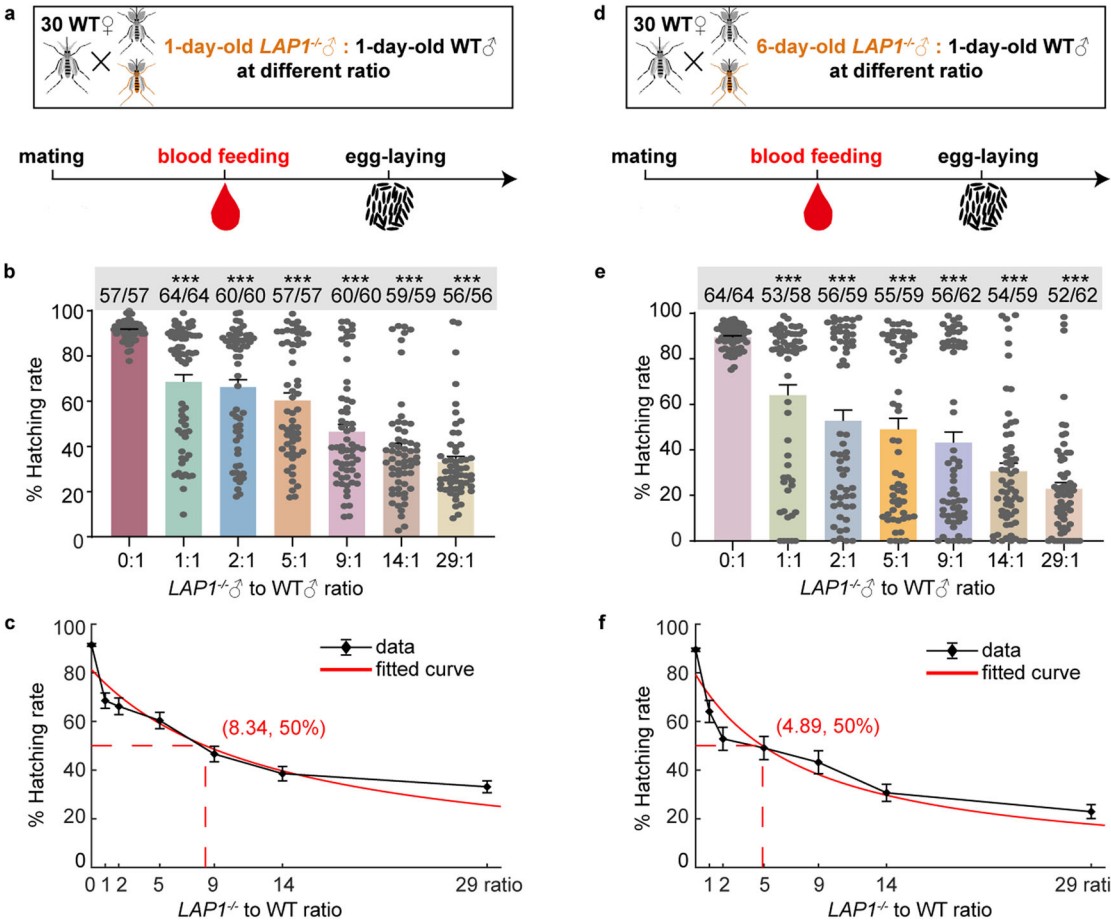

**Fig. 6 | *LAP1⁻ᐟ⁻* males compete with WT males to limit female fertility. a** Scheme of the mating competition assay of 1-day-old male mosquitoes (30 females were paired with 30 males per group). The ratios of *LAP1⁻ᐟ⁻*♂ to WT♂ were 0:1 (0 *LAP1⁻ᐟ⁻*♂ and 30 WT♂), 1:1 (15 *LAP1⁻ᐟ⁻*♂ and 15 WT♂), 2:1 (20 *LAP1⁻ᐟ⁻*♂ and 10 WT♂), 5:1 (25 *LAP1⁻ᐟ⁻*♂ and 5 WT♂), 9:1 (27 *LAP1⁻ᐟ⁻*♂ and 3 WT♂), 14:1 (28 *LAP1⁻ᐟ⁻*♂ and 2 WT♂), and 29:1 (29 *LAP1⁻ᐟ⁻*♂ and 1 WT♂) for a total of 30 males in each group. **b** The fertility of WT females mated with 1-day-old *LAP1⁻ᐟ⁻* and WT males at different ratios was evaluated (*n* = 57 for 0:1; *n* = 64 for 1:1; *n* = 60 for both 2:1 and 9:1; *n* = 57 for 5:1; *n* = 59 for 14:1; *n* = 56 for 29:1). 1-day-old *LAP1⁻ᐟ⁻* and WT males were introduced simultaneously at different ratios (30 males in total) to mate with 30 WT females, and females were given blood meal after 3 days. Egg hatchability was then measured. **c** Curve fitting of the same data from **b**. The dashed line shows the point at which mating with 1-day-old males of indicated *LAP1⁻ᐟ⁻* to WT ratios resulted in a 50% reduction in female fertility. The fitted curve was marked with red. **d** Scheme of mating competition assay of 6-day-old male mosquitoes (30 females were paired with 30 males per group). **e** The fertility of WT females mated with *LAP1⁻ᐟ⁻* and WT males at different ratios. 6-day-old *LAP1⁻ᐟ⁻* and 1-day-old WT males were introduced simultaneously at different ratios (30 males in total) to mate with 30 WT females (*n* = 64 for 0:1; *n* = 58 for 1:1; *n* = 59 for 2:1, 5:1 and 14:1; *n* = 62 for both 9:1 and 29:1). Blood meal was given after 3 days. And egg hatchability was measured. **f** Curve fitting of the same data from **e**. The dashed line shows the point at which mating with 6-day-old males of indicated *LAP1⁻ᐟ⁻* to WT ratios resulted in a 50% reduction in female fertility. The fitted curve was marked with red. Females that died during laying eggs were removed from the study. The top of each column in **b** and **e** shows the number of hatched mosquitoes relative to total mosquitoes. Each dot represents one mosquito. The experiments were independently performed three times with 30 females per experiment. A Kruskal–Wallis test of variance was conducted to detect any significant variation among three replicates, and no significant differences were detected. Then results were combined for further analyses. Statistical significance of hatchability was determined using the Kruskal–Wallis test followed by Dunn's post-hoc tests with Benjamini–Hochberg adjustment for multiple comparisons (**b** and **e**). The *q*-value was calculated by comparing the experimental group with the 0:1 group. Data are shown as mean ± SEM. \**q* < 0.05, \*\**q* < 0.01, \*\*\**q* < 0.001. Source data are provided as a Source Data file.

asthenozoospermia appeared to be associated with defects in sperm mitochondrial membranes[52]. Unlike the findings in other species, we discovered that the 9 + 9 + 1 microtubule structure of sperm dissociated after *LAP1* knockout in mosquitoes (Fig. 4b). Mitochondria were recognized as the organelles for ATP production in response to the energy demands of cells[53]. The autophagy that occurred in *LAP1* mutants was likely the cause of the impairment of mitochondrial derivatives and abnormal structure of microtubules. The autophagy seemed to be one of the reasons for the disappearance of sperm in *LAP1⁻ᐟ⁻* mosquitoes, as shown in Fig. 4b. Although extensive evidence demonstrates that LAP1 plays a function in *Ae. aegypti* spermatogenesis, including the effect on the structure of mitochondrial derivatives and microtubules, further in-depth investigation of details on the molecular mechanism and genes participating in this process is required.

Over the past few decades, SITs have been successful in controlling several insect species, such as the New World screwworm *Cochliomyia hominivorax*, the Mediterranean fruit fly *Ceratitis capitata*, and the Unguja Island *Glossina austeni* tsetse flies[54–56]. Since the 1950s, scientists have successfully applied SIT technology to control mosquito populations[57]. Nevertheless, traditional SIT produces sterile males that possess low mating competitiveness due to mutagenesis of many genes[57]. Therefore, alternative strategies to produce highly competitive sterile males using the CRISPR/Cas9 system or RIDL have become attractive for controlling mosquito populations[58–61]. In our study, we used the CRISPR/Cas9 system to generate low-fertility *LAP1⁻ᐟ⁻* males that show no differences from WT males in terms of physiological parameters under laboratory conditions, including longevity and virus transmission capacity (Supplementary Fig. 5a, b), suggesting that

*LAP1* mutants may be as safe and robust as the WT, but further field experiments are needed to be proved under real field conditions. Moreover, in contrast with the previous SIT in which sterile males could not be maintained as homozygous lines, *LAP1* mutants are able to generate hatchable offspring at an average of 9.5% after the first blood meal (Fig. 2d). Ideally, genetic changes arising from sterility-inducing genes should be transmitted through insect populations without relying on a sustained mass release of sterile insects, which is the concept behind the development of the CRISPR/Cas9 gene editing system[59]. Therefore, mating with *LAP1*[−/−] males is effective in spreading genetic changes to mosquito populations through a low level of fertilization and consequent low number of hatchable offspring. Importantly, the fertilizing ability of sperm in *LAP1*[−/−] males decreased over time (Fig. 5e). Thus, we were able to control populations by mating with *LAP1*[−/−] males at an appropriate age.

Nonetheless, high residual fertility might affect the political and ethical acceptability of this technology, particularly in areas where mosquito-borne diseases are endemic, and it is recommended that it should be kept below 1%[62]. Thus, one of the next steps in the use of *LAP1*[−/−] males for SIT is to overcome the challenge of high residual fertility. Our spermathecal fluid proteome data showed that in addition to LAP1, which was significantly higher in the G2/G1 group, three LAPs (AAEL000424-PA, AAEL001649-PA, and AAEL002978-PA) were observed in this group (Supplementary Data 1). We could study the effect of these LAPs on male sperm and reduce mosquito fertility by double or multiple knockdowns. Most important, transferring laboratory results to practical applications requires further research and work, including investigation of release ratios, environmental conditions, and regulatory strategies, to ensure that the function of population suppression in the field is as effective and safe as that in the laboratory.

Overall, it is conceivable that mosquito populations and vector borne diseases could be reduced by releasing *LAP1*[−/−] males at an indicated age within a certain range that is neither ecologically damaging nor harmful to human health. Moreover, the amino acid sequence of *Ae. aegypti* LAP1 is highly conserved in *Aedes albopictus* and *Culex quinquefasciatus*, with 97% and 91% identities, respectively, indicating that strategies for deleting *LAP1* can be applied to other mosquito species to improve its efficacy in controlling a wide range of mosquito populations. Thus, our work provides a target gene for the gene drive system, further amplifying the function of LAP1 in reducing mosquito populations.

## Methods

### Mosquitoes and virus infection
Liverpool strain *Ae. aegypti* mosquitoes were raised as previously described[36,63,64]. Adult mosquitoes were cultured through supplementing with 10% (wt/vol) sucrose solution and water. The rearing conditions were 28°C, 80% humidity with 12/12-h light/dark cycle. Mosquito strains were maintained by feeding on chicken (4 to 8 weeks old) blood once a month. The ZIKV (MR766 strain; GenBank sequence accession number, HQ234498) and DENV2 (New Guinea C strain; GenBank accession number, M29095) was grown in C6/36 cells (ATCC, CRL-1660) at 28°C with 5% (vol/vol) $CO_2$ in RPMI 1640 medium containing 8% fetal bovine serum (FBS, Cat. 10091148, Invitrogen™, USA) for mosquito oral infection. All viruses and mosquito experiments were conducted under biosafety level 2 (BSL2) conditions and approved by the Bioethics Committee of the Institute of Zoology, Chinese Academy of Science (IOZ-IACUC2020-067).

### SWATH-MS
Spermathecae from 2000 individual female mosquitoes at the virgin (unmated) at 72 h PE, mated at 72 h PE and mated at 5 days PBM stages were pooled for the construction of a spectral library. In addition, spermathecae from 500 individual females at the same time point as

above were collected for analysis. Each group had three replicates. Mass spectrometry was then performed by Beijing Protein Innovation Co., Ltd.

Briefly, the protein is first extracted, enzymatically digested and purified. The enzymatically digested peptides are then subjected to a data-dependent acquisition (DDA) mode to obtain the total ion flow map of the mass spectrometry signal. Peptides were identified using a Triple time-of-flight (TOF) 6600 instrument interfaced with an Agilent 1100 HPLC system (equipped with the ChromXP CL-3 μm,120 Å, 350 μm × 0.5 mm, 1.8 μm 150 μm × 20 cm). To acquire protein identification data, the results got from DDA were retrieved by MaxQuant[65]. In the next step, data were incorporated into Peakview to establish an ion library without the need to import shared peptides[66]. Samples from 500 individual females at the virgin, 72 h PE and 5 days PBM stages were examined using data-dependent acquisition (DIA) mass spectrometry. The detected spectra of the DIA samples were matched, and quantitative information were extracted from the ion library using Peakview. Finally, the protein identification results from Peakview were statistically analyzed. Fold change is the ratio between the mean quantitation values of protein. Statistical significance was determined using a two-sided unpaired *t*-test. The protein with a *p* value below 0.05 and a fold change above 1.5 or below 0.667 was considered differentially expressed.

### RNAi, RNA extraction and qPCR
dsRNA was synthesized using the T7 RiboMAX express RNAi system (Cat.P1320, Promega, USA), and approximately 1.2 μg per 207 nL of dsRNA was microinjected into the thoraxes of 1-day-old female mosquitoes using a Nanoliter 2010 injector (World Precision Instruments, USA). Mosquitoes injected with *EGFP* dsRNA were served as controls. After a 3-day recovery period, they were used for further experiments. The RNAi efficiency of 5-10 mosquitoes was determined at 3 d post-injection. Total RNA of whole mosquitoes or tissue was isolated using TRIzol reagent (Cat.15596018, Invitrogen™, USA). Quantification of viral and *Ribosomal Protein S7* (*rps7*) gene RNA was performed using a one-step SYBR PrimerScript reverse transcription PCR protocol (Cat. RR066A, Takara Bio Inc, Japan) on the Applied Biosystems Step-One Plus system (Applied Biosystems, USA). The parameters were as follows: reverse transcription at 42 °C for 5 min, followed by 40 cycles of 95 °C for 10 s and 60 °C for 30 s. The different samples were normalized to the *Ae. aegypti rps7* gene. Primers used in RNAi and qPCR here were listed in Supplementary Table 3.

### Mosquito lifespan
A group of 30 males were maintained in separate containers and supplemented continuously with water and 10% (wt/vol) sucrose solution. The survival numbers were monitored at an interval of 24 h. Kaplan−Meier (KM) survival curves were plotted using http://www.bioinformatics.com.cn/en, an online platform for data analysis and visualization. The *p*-value was calculated using log-rank test (Mantel−Cox test).

### Mosquito fecundity and fertility
Mating assay as shown in Fig. 2a: Three days after microinjection, 50 females were mated with 50 males for 3 days. Females were then fed blood and transferred to separate oviposition tubes (10-mL centrifuge tube) for egg collection at 3 days PBM. The numbers of eggs were counted at 6 days PBM, and subsequently these eggs were used to calculate the hatchability. Female mosquitoes, dying or unwilling to have blood meal due to microinjection, were eliminated from the experiment.

Mating assay as shown in Fig. 2b−g: Mosquitoes were divided into three groups (WT♂ × WT♀, *LAP1*[−/−]♂ × *LAP1*[−/−]♀, and *LAP1*[−/+]♂ × *LAP1*[−/+]♀) and mated for 3 days. Fifty females were paired with 50 males per group. Females were then allowed to feed blood (the first blood meal) and moved to separate oviposition tubes for egg collection at 3 days

PBM. After 3 days of oviposition, surviving females were transferred to a new container with water and 10% (w/v) sucrose for further blood meal[67]. Eggs from the first blood meal were counted at 6 days PBM and subsequently these eggs were used to calculate hatchability. After three days of recovery, surviving females were given a blood feeding again (the second blood meal). They were then placed in oviposition tubes to lay eggs at 3 days PBM, and the numbers of eggs were counted at 6 days PBM. These eggs were used to calculate hatchability.

Mating assay as shown in Fig. 2h–m: Mosquitoes were separated into three groups (WT♂ × WT♀, $LAP1^{-/-}$♂ × WT♀, and WT♂×$LAP1^{-/-}$♀) and mated for 3 days. Fifty females were paired with 50 males in each group. Egg deposition and hatchability were assessed after the first and second blood meals using the same mating assay as shown in Fig. 2b–g.

### Generation of *LAP1* mutant by CRISPR/Cas9

Three sgRNAs targeting the third exon of *Ae. aegypti LAP1* gene were designed using the website (https://zlab.bio/guide-design-resources), and T7 sequence was added at the 5′ end as an upstream primer. Downstream primer was the same as that reported in a previous paper[68]. All sgRNAs were amplified using PCR and transcribed in vitro using the T7 RiboMAX™ Express RNAi System (Cat.P1320, Promega, USA). sgRNAs (50 ng/µL) and cas 9 protein (333 ng/µL, Cat.CP01-50, PNABIO, China) were mixed and incubated at 37 °C for 20 min before injection ($n = 600$). After a 5-day recovery period, eggs were hatched and the mutation was tested using PCR and DNA sequencing. The *LAP1* mutant line was backcrossed to the WT line for three generations. The *LAP1* strain is maintained as homozygotes. Primers used here were shown in Supplementary Table 3.

### Western Blot

WT and *LAP1* mutant mosquitoes were collected at G2 and extracted in RIPA lysis buffer (Cat. CW2333S, CWBIO, China) with 1 × protease inhibitor (Cat. 78425, Pierce, USA). Samples were separated using a precast gel (Cat. BE6929-11, EASYBIO, China) and were transferred to PVDF membranes (Cat. IPVH00010, Millipore, USA). After blocking, the membrane was incubated with mouse anti-LAP1 antibody primary antibody (1:5000) and conjugated anti-mouse IgG (H&L)-HRP secondary antibody (1:10000) (Cat.BE0102-100, EASYBIO, China) overnight at 4 °C. The LAP1 mouse polyclonal antibody was purified at Beijing Protein Innovation Co., Ltd. Mouse monoclonal β-actin antibody (1:5000) (Cat. BE0033-100, EASYBIO, China) was served as the loading control. Images were visualized using SuperSignal West Pico Substrate (Cat. 34577, Pierce, USA) and the blots were scanned from Image Lab software (Image Lab 5.2) by ChemiDoc XRS⁺ System (Bio-Rad, USA).

### Immunofluorescence assay and confocal microscopy

Sperm isolated from testes were incubated in Lab-TekII Chamber Slide (Cat. 154534, Thermo Fisher Scientific, USA) with RPMI 1640 medium for 30 min to adhere and then stained in MitoTracker Red CMXRos (0.5 µM, Cat.M7512-50, Thermo Fisher Scientific, USA) for 30 min. After fixation in 4% paraformaldehyde (Cat. E672002-0500, BBI, China) for 30 min and immersion in 0.5% Triton X-100 for 10 min, sperm were blocked in 3% BSA solution for 1 h at room temperature. Subsequently, they were stained with mouse anti-LAP1 antibody (1:500) and then Alexa Fluor 488 conjugated anti-mouse secondary antibody (1:2000) (Cat. A11001, Invitrogen™, USA). Hoechst 33258 (Cat. H3569, Invitrogen™, USA) was used to stain the nucleus at a final concentration of 2 µg/mL. Fluorescence was captured by Zeiss Zen 2010 software (version 6.0, Zeiss, Germany) using a confocal microscope (Zeiss LSM 710, Germany). For the immunofluorescence of spermatheca, tissue adhered to Lab-TekII Chamber Slide (Cat. 154534, Thermo Fisher Scientific, USA). The methods of fixation, permeabilization, blocking, and incubation were the same as above.

### Transmission electron microscopy

Testes were fixed with 2.5% (vol/vol) glutaraldehyde and 0.1% tannic acid in phosphate buffer (PB) (0.1 M, pH 7.4), washed twice in PB and twice in ddH₂O. Then the tissue was first immersed in 1% (wt/vol) OsO4 and 1.5% (wt/vol) potassium ferricyanide aqueous solution at 4 °C for 2 h. After washing, the tissue was dehydrated through graded alcohol (30%, 50%, 70%, 80%, 90%, 100%, 100%, 10 min each) into pure acetone (2 × 10 min). Tissue was infiltrated in graded mixture (3:1, 1:1, 1:3) of acetone and SPI-PON812 resin (21 mL SPI-PON812, 13 mL DDSA and 11 mL NMA) and then changed to pure resin. Finally, the tissue was embedded in pure resin with 1.5% BDMA and polymerized at 45 °C for 12 h and at 60 °C for 48 h. The ultrathin sections (70 nm thick) were cut with a microtome (Leica EM UC6, Germany), double stained with uranyl acetate and lead citrate. TEM was conducted on a FEI Tecnai G2 F20 TWIN TMP instrument at 200 kV with magnifications as indicated.

### Detection for suppression of female fertility by allowing WT females that mated with $LAP1^{-/-}$ males to feed on blood at indicated days

After $LAP1^{-/-}$ males mated with a total of 30 WT virgin females for 3 days, the males were removed and the females were allowed to feed blood at 0, 4, 8, 12, and 16 days after mating. Then, individual females were transferred to a separate tube for oviposition, egg collections, and the hatchability calculation (the numbers of hatched larvae/total number of eggs from individual females) using the same method as in the "Mosquito fecundity and fertility" section. Females that died during laying eggs were removed from the calculation. Three independent assays were performed per condition.

### Detection for suppression of female fertility by mating with indicated day-old WT and $LAP1^{-/-}$ males

A total of 30 WT virgin females mated with indicated day-old (1-, 3-, 5-, 7-, 9-, and 11-day-old males) WT and $LAP1^{-/-}$ males for 3 days before the blood meal. After a blood meal, individual mosquitoes were transferred to separate tubes for egg collections. The numbers of hatched larvae from individual mosquitoes were counted using the same method as in the "Mosquito fecundity and fertility" section. Females that did not take a blood meal and died during laying eggs were removed from the study. Per condition was repeated at least three times.

### Mating competition assay

1-day-old or 6-day-old $LAP1^{-/-}$ and 1-day-old WT males were introduced simultaneously at different ratios (30 males in total) to mate with a total of 30 WT virgin females for 3 days. $LAP1^{-/-}$♂ to WT ♂ ratio was 0:1, 1:1, 2:1, 5:1, 9:1, 14:1, and 29:1. Then, females were allowed to take a blood meal. After 3 days, individual mosquitoes were transferred to separate tubes for egg collections. The hatching rate (the number of hatched larvae relative to the total number of eggs from individual female) were calculated. Females that did not suck blood and died during laying eggs were removed from the study. Each condition was repeated 3 times.

### Curve fitting of experimental data for suppression of female fertility by $LAP1^{-/-}$ males

The maximum likelihood estimates of the curve parameters in Figs. 5f and 6f were performed as follows. Under the experimental condition $i$ and biological repeat $j$, the hatching rate is $h_{ij}(\theta)$ and we considered all female were mated 3 days after the mixing of males and females.

In our experiments, we added the $LAP1^{-/-}$ and WT males simultaneously at 0 days and 6 days after emergence, then calculated the hatchability and fitted it with the following equation:

$$h_{ij}(\theta) = \frac{h_c}{1 + \frac{K_c N_{LAP1,i}}{N_{WT,i}}} \times 100\% \qquad (1)$$

Here, the parameter $\theta$ contains $h_c$ and $K_c$. $h_c$ is the maximum hatchability under all conditions. $K_c$ is the sperm motility of $LAP1^{-/-}$ males normalized to WT males. $N_{LAP1,i}$ and $N_{WT,i}$ are respectively the number of $LAP1^{-/-}$ males and WT males under experimental condition $i$.

We had verified that $LAP1^{-/-}$ males have almost the same competitiveness as the WT males. Therefore, only $\frac{N_{WT}}{N_{WT}+K_c N_{LAP1}}$ of the eggs were alive. We estimated $h_c$ and $K_c$ using the maximum likelihood estimation. The values of males 0 days after emergence are $h_c = 0.813 \pm 0.041(95\%CI)$ and $K_c = 0.075 \pm 0.016(95\%CI)$. The values of males 6 days after emergence are $h_c = 0.792 \pm 0.059(95\%CI)$ and $K_c = 0.120 \pm 0.033(95\%CI)$.

In the experiment that we added the $LAP1^{-/-}$ of different days after emergence and WT females, we got the hatchability and fitted the curve with the following equation:

$$h_{ij}(\theta) = \frac{h_c}{1 + E_c(t_i - 1)} \times 100\% \qquad (2)$$

Here, the parameter $\theta$ contains $h_c$ and $E_c$. $h_c$ is the maximum hatchability under all conditions. $E_c$ is the influence of time to the sperm motility. $t_i$ is the days after emergence of males under experimental condition $i$. We assumed males reach sexual maturity 1 day after emergence. We estimated $h_c$ and $E_c$ using the maximum likelihood estimation. The values are $h_c = 0.190 \pm 0.032(95\%CI)$ and $E_c = 0.192 \pm 0.095(95\%CI)$.

In the experiment that WT females were allowed to give a blood meal in different days after mixing the $LAP1^{-/-}$ males, we detected hatching rates and use the following formula to fix it:

$$h_{ij}(\theta) = \frac{h_c}{1 + E_c t_i} \times 100\% \qquad (3)$$

Here, the parameter $\theta$ contains $h_c$ and Ec. $h_c$ is the maximum hatching rate under all conditions. $E_c$ is the influence of time to the sperm motility. $t_i$ is the blood-meal day after 3 days of mating with $LAP1^{-/-}$ males and WT females under experimental condition $i$. We estimated $h_c$ and $E_c$ using the maximum likelihood estimation. The values are $h_c = 0.137 \pm 0.020(95\%CI)$ and $E_c = 0.426 \pm 0.210(95\%CI)$.

### Fried's Competitiveness Index for suppressing female fertility in $LAP1^{-/-}$ males

Fried's competitiveness index was used to calculate the mating competitiveness of $LAP1^{-/-}$ males in blood-feeding experiments on females for indicated days and mating experiments between females and indicated age mutants. In the competitiveness index developed by Fried, it assumes that the proportion of hatching eggs ($H_m$) is determined from a mixture of $S$ (incompatible) and $W$ (WT) insects as follows:

$$H_m = \frac{W H_w}{W + S} + \frac{S H_s}{W + S} \qquad (4)$$

$H_w$ and $H_s$ represent the expected proportions of egg hatching, when WT females are exposed to WT males or incompatible males, respectively. The underlying equation assumes the equal mating competitiveness between WT and incompatible males. However, it is more reasonable to assume unequal competitiveness, thus leading to a revised equation as follows:

$$H_m = \frac{W H_w}{W + cS} + \frac{cS H_s}{W + cS} \qquad (5)$$

$C$ represents the coefficient of mating competitiveness. The parameter $c$ can be regarded as the probability, $\frac{P_s}{(1-P_s)}$, which means a female will mate with an incompatible male rather than a WT male, and

both are in equal numbers (i.e., W = S). And $P_s$ represents the probability that a mated female will choose to mate with an incompatible male rather than a WT male. The above equation could be rearranged to acquire to yield $c$. Thus, we exposed WT females to WT males, $LAP1^{-/-}$ males, and 1:1 mixtures of WT and $LAP1^{-/-}$ males to acquire $H_{WT,i}$, $H_{LAP1,i}$, and $H_{m,i}$, which are the hatching rates in condition $i$, and calculated $c_i$ in condition $i$ using the equation

$$c = \frac{W(H_w - H_m)}{S(H_m - H_s)} \qquad (6)$$

Thus, we exposed WT females to WT males, $LAP1^{-/-}$ males, and 1:1 mixtures of WT and $LAP1^{-/-}$ males to obtain $H_{WT,i}$, $H_{LAP1,i}$, and $H_{m,i}$, the proportions of eggs that hatched in condition $i$, and calculated $c_i$ in condition $i$ using the equation

$$c_i = \frac{H_{WT,i} - H_{m,i}}{H_{m,i} - H_{LAP1,i}} \qquad (7)$$

### Viral infection

Seventy females from the $LAP1^{-/-}\male \times LAP1^{-/-}\female$, $LAP1^{-/+}\male \times LAP1^{-/+}\female$, and WT$\male \times$ WT$\female$ groups were mated with males at a 1:1 ratio. Three- to four-day-old WT, $LAP1^{-/+}$ and $LAP1^{-/-}$ female mosquitoes were starved for 15–18 h before viral infection. Mouse blood containing $1.0 \times 10^6$ FFU/mL ZIKV or DENV was preheated for 30 min at 37 °C. The mixture was then fed to females for 30 min at 37 °C through an in vitro membrane feeding apparatus. Mosquitoes were anesthetized for 15 min at 4 °C, and females with a full-blood meal were transferred to new containers. The mosquitoes were then reared in an incubator at 28 °C, 80% humidity, and a 12/12-h light-dark cycle. Total RNA was extracted from a single mosquito at 3-, 7-, and 10-days post infection, and the viral RNA level was measured using qPCR. Primers used in the virus detection were shown in Supplementary Table 3.

### Statistical analysis

Proteomic data were performed using http://www.bioinformatics. com.cn, which is used for analysis and visualization (Fig. 1a–c, Supplementary Fig. 1b, c, and Fig. 5a). Volcano plot analysis in Fig. 1a, GO enrichment analysis in Supplementary Fig. 1b, and KEGG functional classification in Supplementary Fig. 1c were performed by enhanced-volcano R package (1.13.2), Goplot R package (1.0.2), and circlize R package (0.4.15) in the R environment, respectively. The Kaplan−Meier curves in Supplementary Fig. 5c were plotted using survival (3.5-7) and survminer R package (0.4.9), and the $p$-value was calculated using log-rank test (Mantel−Cox test). The DAVID web server (https://david. ncifcrf.gov/) was used for GO enrichment analysis and the $p$-value was determined by a two-sided Fisher's exact test with Benjamini−Hochberg adjustment. KEGG enrichment analysis was performed using KOBAS web server (http://bioinfo.org/kobas/) and the $p$-value was tested by a two-sided Fisher's exact test with Benjamini−Hochberg adjustment. Venn diagram in Fig. 1b and Sankey dot plot in Fig. 1c were performed using the matplotlib python package (3.7.0). Statistical analyses of the remaining data were done by GraphPad Prism statistical software. Data are shown as mean ± SEM. Statistical significance is provided in the figure legends. Statistical significance of egg deposition and hatchability in Figs. 2 and 6 was determined using the Kruskal−Wallis test followed by Dunn's post-hoc tests with Benjamini−Hochberg adjustment for multiple comparisons. A two-sided Fisher's exact test was used to statistically analyze the ratio of females producing hatchable eggs to total mosquitoes. Statistical significance of the mRNA abundance of $LAP1$ in Fig. 3a were determined using the one-way ANOVA test followed by Tukey's post hoc tests with Benjamini−Hochberg adjustment for multiple comparisons. Statistical significance of the expression of $LAP1$ in Fig. 3d and

hatchability in Fig. 5 was determined using the two-sided multiple Mann–Whitney test with Benjamini–Hochberg adjustment. The *p*-value of the length of the ovarian follicles in Fig. 3c was determined by two-sided Mann–Whitney test. For curve fitting, all error bars represent ± SEM (Figs. 5c, f and 6c, f), which is 95% CI to indicate that the fitted curves are in the range of all error bars.

## Reporting summary
Further information on research design is available in the Nature Portfolio Reporting Summary linked to this article.

## Data availability
The mass spectrometry proteomics data generated in this study have been deposited in the ProteomeXchange Consortium (http://proteomecentral.proteomexchange.org) via the iProX partner repository[69,70] under accession number PXD039869. All data, supplementary information, and Source Data files are available in this article. Source data are provided with this paper.

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

## Acknowledgements

This work was supported by National Key Plan for Scientific Research and Development of China No. 2021YFC2600100 and 2019YFC1200504, and National Natural Science Foundation of China Grants 32370522 and 32370518. We thank Zhongshuang Lyu and Xixia Li for assistance with electron microscopy sample preparation and TEM imaging at the Center for Biological Imaging (CBI), Institute of Biophysics, Chinese Academy of Science.

## Author contributions

X.S., X.W., and Z.Z. designed the study; X.S. performed assay and analyzed data; X.S., X.W., K.S., X.L., J.S., A.S.R., and Z.Z. contributed to reagents/materials/analysis tools. X.S., X.W., A.S.R., and Z.Z. wrote the manuscript.

## Competing interests

The authors declare no competing interests.
