## [Peer Review File · Nature Communications]

Leucine aminopeptidase1 controls egg deposition and hatchability in male *Aedes aegypti* mosquitoesReviewers' Comments:

Reviewer #1:

Remarks to the Author:

In this MS, Sun et al identified LAP1 in *Aedes aegypti* males with the potential to be used for population suppression. They have provided a comprehensive story to show the discovery, functions of LAP1 in both males and females, the mechanism for inducing sterility in males, and the ability of knockout males to induce sterility for population suppression. Given the significant worldwide effort to develop SIT for mosquito-borne disease control, this study provides a new angle to move this field forward. The experiments are well designed with solid results to illustrate the role of LAP1 in *Aedes aegypti* reproduction. Below are some comments for authors to further improve the MS.

Authors are suggested to compare the number of sperms in male testis and female spermatheca between the mutant and wild-type mosquito, which can contribute to the female sterility induced by LAP1 $-/-$ males.

To better clarify the role of LAP1 in the knockout line, authors may need an experiment to supplement LAP1 using recombinant plasmid to test whether it can rescue the sterility induced by LAP1 knockout. This can exclude the effect of any non-target mutation induced by CRISPAR/cas9 when generating the knockout line.

In figure 5, the authors are suggested to calculate the male mating competitiveness of LAP1 $-/-$ at different ages and different release ratios (better using the data with ratios less than 4), using Fried's Competitiveness Index, which is commonly used in the field of SIT. Although aged LAP1 $-/-$ males induced better sterility, their mating competitiveness index may be too low (if less than 0.2) to be useful for population suppression. The current Curve fitting of experimental data is not straightforward and provides very vague information on the male mating quality.

In the discussion, the authors should acknowledge the residual fertility of LAP1 $-/-$ males are too high for them to be used in SIT in the field at this stage, which normally needs to be around (or less than) 1% according to the guideline for the field use of SIT by IAEA, even 9- and 11-day-old LAP1 $-/-$ males were used (6.2% and 4.9%) or 7.2% after the first bloodmeal is considered. This will not affect the significance of this work, but can provide a direction for future studies to improve the LAP1-based technology development.

Line 59-62, the statement has a grammar issue.

Line 68, investigation of their reproduction, what does "their reproduction" mean here?

Line 215, it is mainly expressed in sperm. It should be testis, rather than sperm. Otherwise, the expression should also be observed in the spermatheca.

Line 419-421, there is no male flight ability test in the supplemental data. Instead, the sound-seeking assay is included, which is very unclear why this assay is performed, what the results mean, and how significant these results are related to the population suppression in the field as claimed in the discussion.

There is no legend for the supplemental figures.

Reviewer #2:

Remarks to the Author:

Summary and Recommendation

In this study Sun and colleagues investigate the role of protein LAP1 in *Aedes aegypti* reproduction. Females with reduced or ablated LAP1 proteins produced through RNAi and CRISPR, lay significantly less eggs and those eggs are less likely to hatch. They also uncover an apparent role of LAP1 in male physiology. Not only do they find that gene encoding LAP1 is highly expressed in sperm, they also use crosses with LAP1 deficient CRISPR lines to show that females mating with males lacking in LAP1 are less fecund and fertile. Interestingly, this effect on fertility is age dependent. Matings with older LAP1 deficient males result in smaller proportions of eggs that hatch from matings. Finally the study presents a set of experiments designed to test the competitive ability and safety of a LAP1 knockout line.

The data presented are exciting and the SWATH-MS, and physiological aspects in particular are very much worthy of publication. However, there are critical details about the methodology and analysis missing from the manuscript that prevents us from recommending the manuscript for publication in its current form. We have additional concerns about the interpretation of some of the findings. Overall, the study reveals exciting aspects of basic mosquito physiology and reproductive biology. We would recommend focusing on these aspects instead of extrapolating into the applicability of these findings for mass releases. We found these to be unconvincing and they do a disservice to the rest. We have detailed these major concerns below and provided a list of minor concerns which the authors may wish to address.

Major Concerns:

Missing details in methods:

There are no methods provided (at least in what we were able to find as reviewers) for the experiments reported in Figure 1 A, C, D.

The sample sizes for mating experiments as described in the methods and results do not match the figures. For example for the initial crosses of homozygous and heterozygous LAP1 knockouts: The methods indicate that 30 females were held with some unknown number of males for 3 days. After which individual females were separated into tubes and allowed to oviposit. It was from these females that the fecundity and fertility data was measured. Three independent assays were performed per condition. From this we would expect that for each treatment we would have data for ~90 females total. (Lines 491-501)

The written results indicate that there was a significant reduction in hatchability with no statistical information and refers the reader to Figure 2b (Lines 184-188).

Figure 2b The sample sizes appear for each treatment range from 18-36. Where did the rest of the females go? It appears as though a significant proportion (~2/3) have died or not during the course of the study. This all needs to be explained clearly.

This is true of all of the mating assays. The reader must be able to determine how the reported sample size in the results have come from the methods as described.

In the Data availability section it indicates that all the data underlying these figures is in a Source Data file. We were not able to access this as a reviewer to confirm sample sizes.

Clarification and justification of statistical analysis:

It is unclear what statistical tests have been applied to the data. It appears that Mann-Whitney tests were used on egg deposition and hatchability data (Lines 646-647). On Line 960 it indicates that a Fisher's exact test was used for hatchability data in some experiments, but not in others.

It looks as though a series of pairwise comparisons have been made between the control in a given experiment and each experimental treatment. This type of pairwise comparison without appropriate correction can inflate significance. Further, this approach does not account for the fact that individual females measures are nested within the replicate cage.

The authors need to clarify the statistical analysis for each experiment. Normally, we would recommend using a generalized linear mixed model or similar analysis to account for nested experimental design.

Explanation and rationale for sound seeking assay:

The sound seeking assays (Lines 622-644) lack clear rationale and description.

The dimensions of this box likely inhibit free flight (4 cm)

The definition of metrics such as 'large jump' and 'trajectory birth' are missing along with an explanation of what these metrics indicate about male behavior.

Please clarify how this relates to male mating behaviour

We were unable to find any statistical analysis of the flight assay.

Details and interpretation of viral infection data

Critical experimental details missing here including number of females fed and temperature at which females were held

Viral loads are measured at a single time point which may not be representative of infection dynamics (Line 640).

No difference in infection load at a single temperature and time point is not sufficient evidence to support the statement on Line 340-344 about safety. At best these experiments are preliminary and either should be expanded or removed from the manuscript.

Interpretation of competitive mating assays.

Our understanding of the data presented in Figure 5 is that LAP1 $-/-$ males allowed to mate with WT females in the absence of competition result in extremely low hatchability. The curve suggests that a female mating with a 3.8398 day old LAP1 $-/-$ male would be predicted to have a 10% hatch rate and that this rate would continue to fall as female mates with older males. [NOTE: methods do not clarify how many males are held with the 30 females.]

In Figure 6, the reported hatch rate when 29 LAP1 $-/-$ males compete with 1 WT male is still >20%. This suggests that the WT male is mating a disproportionate number of females and the LAP1 $-/-$ males are highly uncompetitive. This may be an issue with the details of these assays not being clear, but certainly needs clarification. [NOTE: Figure 6 c and f label the x axes with 'days' rather than presumably 'ratio']

Further in these competitive assays, the ratio of LAP1 $-/-$ males to WT males is varied, but the intensity of competition for females overall remains constant because the total number of males to females is equal across the assays. This ratio does not represent the operational sex ratio in the field. This point needs to be addressed.

Minor Concerns:

1. Author ASR is not mentioned in the Author Contributions. The contribution of this author to the work needs to be specified.
2. It is unclear why the authors choose to focus exclusively on male physiology. The justification provided is that the phenotype is more pronounced in the matings between LAP1 $-/-$ males and WT females, but there is still a significant difference in hatchability and egg deposition in the first gonotrophic cycle in the matings involving WT males and LAP1 $-/-$ females suggesting an important role in female reproduction as well. Some extended discussion of this interesting result is warranted.
3. Why were females not provided a bloodmeal on the same day as their matings? Wouldn't this be the most natural scenario? The earliest females receive these in Figure 5 and 6 experiments is 3 days post mating.
4. Figure 2 - For Hatching rate (middle panels Part B and C) the y axis should stop at 100%. The use of different colours for each bloodmeal are unnecessary. We suggest single colors would be clearer. There is small typo in the y-axis for the right panels in BC should read hatched/total mosquito
5. Figure 3- For Part A please display the individual biological replicates (as in Part C). For Part D there is not a clear description where exactly this data have come from or the meaning of the y-axis.
6. Figure 5- the legend discusses sperm motility but that was not measured here. We believe fertility may be a more appropriate term, or simply using hatchability.
7. Figure 6- Typo in x-axis of parts C and F- remove "days", Here again, difficult to judge due to lack of detail, but the error bars do not appear to vary with the spread of the data. This could just be visual

thinking, but would be good to confirm these are accurate?
Line 470 in SWATH-MS methods: the acronym DIA is not explained.
Provide references for all software and programmes- e.g. Peakview

Reviewer #3:
None

REVIEWER COMMENTS

Reviewer #1 (Remarks to the Author):

1. In this MS, Sun et al identified LAP1 in *Aedes aegypti* males with the potential to be used for population suppression. They have provided a comprehensive story to show the discovery, functions of LAP1 in both males and females, the mechanism for inducing sterility in males, and the ability of knockout males to induce sterility for population suppression. Given the significant worldwide effort to develop SIT for mosquito-borne disease control, this study provides a new angle to move this field forward. The experiments are well designed with solid results to illustrate the role of LAP1 in *Aedes aegypti* reproduction. Below are some comments for authors to further improve the MS.

Authors are suggested to compare the number of sperms in male testis and female spermatheca between the mutant and wild-type mosquito, which can contribute to the female sterility induced by LAP1-/- males.

Answer:

Thank you for pointing this out. We previously considered whether mutants could alter sperm counts to induce female infertility and performed related experiments. However, due to a large number of mosquito sperm, sperm bundles and error-prone sampling, it is not possible to accurately count sperm (Shaw, W. R., et al. "Mating activates the heme peroxidase HPX15 in the sperm storage organ to ensure fertility in *Anopheles gambiae*." *Proceedings of the National Academy of Sciences* 111.16(2014).). Previous studies have shown that the seminal vesicles of wild-type *Ae. aegypti* males were filled with sperm using a Zeiss Axio Zoom.V16 stereomicroscope, but the β 2-tubulin (*B2t*) mutants were devoid of spermatozoa, explaining the basis of male sterility in the mutant (Chen, Jieyan, et al. "Suppression of female fertility in *Aedes aegypti* with a CRISPR-targeted male-sterile mutation." *Proceedings of the National Academy of Sciences* 118.22 (2021): e2105075118.). However, in our study, both WT and mutant spermathecae were full of spermatozoa and we were unable to determine whether they differed, so this result is not shown in the article (Figure below).

Figure. Comparison of sperm between WT♂ × WT♀ and *LAP1*^{-/-}♂ × *LAP1*^{-/-}♀ groups. Images were captured using a Zeiss Axio Observer A1 stereomicroscope at 40 × magnification (scale bar: 100 µm). Black arrows: sperm.

2. To better clarify the role of LAP1 in the knockout line, authors may need an experiment to supplement LAP1 using recombinant plasmid to test whether it can rescue the sterility induced by LAP1 knockout. This can exclude the effect of any non-target mutation induced by CRISPR/cas9 when generating the knockout line.

Answer:

We agree with the reviewer in that availability of CRISPR/Cas9-mediated knock-in approaches for mosquito would be ideal. However, mutants were acquired for nearly 6 months because CRISPR/Cas9-mediated knock-out of *LAP1* affects the fertility of female mosquitoes. Furthermore, previous studies have shown that the success rate of germline CRISPR/Cas9-mediated knock-in mutant mosquitoes is approximately 0.2%, and the efficiency of achieving secondary editing on top of the mutant would be even lower (Li HH, Li JC, Su MP, Liu KL, Chen CH. Generating mutant *Aedes aegypti* mosquitoes using the CRISPR/Cas9 system. STAR Protoc. 2021 Apr 8;2(2):100432. doi: 10.1016/j.xpro.2021.100432. PMID: 33899015; PMCID: PMC8058568.). Importantly, our lab is currently unable to successfully construct mutants using

CRISPR/Cas9-mediated knock-in approaches in a short time. Therefore, under the current circumstances, we used alternative approaches to demonstrate the function of *LAPI* and to validate the off-target effect. Firstly, studies have shown that no matter the sgRNA design, it is not possible to fully rule out the possibility of off-target effects in *Ae. aegypti* because both genome assembly and annotation are incomplete (Li HH, Li JC, Su MP, Liu KL, Chen CH. Generating mutant *Aedes aegypti* mosquitoes using the CRISPR/Cas9 system. STAR Protoc. 2021 Apr 8;2(2):100432. doi: 10.1016/j.xpro.2021.100432. PMID: 33899015; PMCID: PMC8058568). All of the genotype confirmed that *LAPI* mutant lines were therefore back crossed with the wild-type strain for three generations to eliminate the chance of off-target effects. Secondly, online tools, such as *Cas-OFFinder* (Bae S, Park J, Kim JS. Cas-OFFinder: a fast and versatile algorithm that searches for potential off-target sites of Cas9 RNA-guided endonucleases. Bioinformatics. 2014 May 15;30(10):1473-5. doi: 10.1093/bioinformatics/btu048. Epub 2014 Jan 24. PMID: 24463181; PMCID: PMC4016707) and *CHOP-CHOP*, that allow us to predict off-target sites based on similarity, likelihood of cleavage based on the position of the mismatches, and number of mismatches. Results from two online tools showed the experimental sgRNA (GAGCTTGAACAGCTGGACGA) without any potential off-targets. Finally, to further confirm the function of *LAPI*, we designed three types of dsRNA (the full length of *LAPI* is 1608 bp, and the coverage of *LAPI*-1, *LAPI*-2, and *LAPI*-3 of 676 to 1097 bp, 1064 to 1556 bp, and 122 to 694 bp, respectively) for knocking it down in virgin males. After a 3-day recovery period, these male mosquitoes were mated with females for 3 days. Next, egg deposition and hatchability were assessed after the first blood meal (Figure below). Results showed that fertility (hatchability) was suppressed in females mated with *LAPI*-1, *LAPI*-2, and *LAPI*-3 dsRNA-treated male mosquitoes compared with those mated with *EGFP* dsRNA-treated males. These experiments clarified the precise role of *LAPI* in the knockout line and excluded the possibility of off-target.

Figure. Analysis of the effect of *LAP1* on male fertility. dsRNA was microinjected into the thorax of 1-day-old virgin males (n=50). After a 3-day recovery period, these mosquitoes were mated with 30 females for 3 days. Mosquitoes injected with *EGFP* dsRNA were served as controls. The top of each column shows the number of hatched mosquitoes relative to total mosquitoes. Statistical significance was determined using a one-way ANOVA with multiple comparisons.

3. In figure 5, the authors are suggested to calculate the male mating competitiveness of *LAP1*^{-/-} at different ages and different release ratios (better using the data with ratios less than 4), using Fried's Competitiveness Index, which is commonly used in the field of SIT. Although aged *LAP1*^{-/-} males induced better sterility, their mating competitiveness index may be too low (if less than 0.2) to be useful for population suppression. The current Curve fitting of experimental data is not straightforward and provides very vague information on the male mating quality.

Answer:

These are excellent suggestions and we have corrected as suggested. We added in the result and methods such as below.

“Fried's Competitiveness Index was used to quantify the mating competitiveness of *LAP1*^{-/-} males in blood-feeding experiments on females for indicated days (0, 4, 8, 12, and 16 days after mating), and it was 0.95, 0.98, 1.09, 1.05, and 1.07, respectively, indicating that the mating competitiveness of *LAP1*^{-/-}♂ and WT♂ was comparable.”

“We also measured the mating competitiveness of *LAP1*^{-/-} males in mating experiments between females and indicated age mutants using Fried's Competitiveness Index, which was 1.09, 1.83, 1.973, 2, 1.68, and 1.87 for mating with 1-, 3-, 5-, 7-, 9-, and 11-day-old males, respectively, indicating that *LAP1*^{-/-}♂ were comparable to or more competitive than WT♂ for mating.” in Result part section “*LAP1* mutation leads to the sperm reduced fertility” .

4. In the discussion, the authors should acknowledge the residual fertility of *LAP1*^{-/-} males are too high for them to be used in SIT in the field at this stage, which normally needs to be around (or less than) 1% according to the guideline for the field use of SIT by IAEA, even 9- and 11-day-old *LAP1*^{-/-} males were used (6.2% and 4.9%) or 7.2% after the first bloodmeal is considered. This will not affect the significance of this work, but can provide a direction for future studies to improve the *LAP1*-based technology development.

Answer:

Thank you so much for the comment. We added the following in the discussion.

Nonetheless, high residual fertility might affect the political and ethical acceptability of this technology, particularly in areas where mosquito-borne diseases are endemic, and it is recommended that it be kept below 1%⁶³. Thus, one of the next steps in the use of *LAP1*^{-/-} males for SIT is to overcome the challenge of high residual fertility. Our spermathecal fluid proteome data showed that in addition to *LAP1*, which was significantly higher in the G2/G1 group, three *LAPs* (AAEL000424-PA, AAEL001649-PA, and AAEL002978-PA) were observed in this group (Supplementary Table 1). We could study the effect of these *LAPs* on male sperm and reduce mosquito fertility by double or multiple knockdowns.

Further details are provided in the methods below:

“Fried's competitiveness index was used to calculate the mating competitiveness of *LAP1*^{-/-} males in blood-feeding experiments on females for indicated days and mating experiments between females and indicated age mutants. In the competitiveness index

developed by Fried, it assumes that the proportion of hatching eggs (H_m) is determined from a mixture of S (incompatible) and W (WT) insects as follows:

$$H_m = \frac{WH_w}{W+S} + \frac{SH_s}{W+S}$$

H_w and H_s represent the expected proportions of egg hatching, when WT females are exposed to WT males or incompatible males, respectively. The underlying equation assumes the equal mating competitiveness between WT and incompatible males. However, it is more reasonable to assume unequal competitiveness, thus leading to a revised equation as follows:

$$H_m = \frac{WH_w}{W+cS} + \frac{cSH_s}{W+cS}$$

C represents the coefficient of mating competitiveness. The parameter c can be regarded as the probability, $\frac{P_s}{\{1-P_s\}}$, which means a female will mate with an incompatible male rather than a WT male, and both are in equal numbers (i.e., $W = S$). And P_s represents the probability that a mated female will choose to mate with an incompatible male rather than a WT male. The above equation could be rearranged to acquire to yield c . Thus, we exposed WT females to WT males, $LAPI^{-/-}$ males, and 1:1 mixtures of WT and $LAPI^{-/-}$ males to acquire $H_{WT,i}$, $H_{LAP1,i}$, and $H_{m,i}$, which are the hatching rates in condition i , and calculated c_i in condition i using the equation

$$c = \frac{W(H_w - H_m)}{S(H_m - H_s)}$$

Thus, we exposed WT females to WT males, $LAPI^{-/-}$ males, and 1:1 mixtures of WT and $LAPI^{-/-}$ males to obtain $H_{WT,i}$, $H_{LAP1,i}$, and $H_{m,i}$, the proportions of eggs that hatched in condition i , and calculated c_i in condition i using the equation

$$c_i = \frac{H_{WT,i} - H_{m,i}}{H_{m,i} - H_{LAP1,i}}$$

” in Methods part section “Fried's Competitiveness Index for suppressing female fertility in $LAPI^{-/-}$ males”.

5. Line 59-62, the statement has a grammar issue. In *Ae. aegypti*, precision-guided sterile insect technique (pgSIT) was developed using a CRISPR-based approach to generate flightless females and sterile

males 7.

Answer:

Thanks. We have corrected this as “In *Ae. aegypti*, it has developed a precision-guided sterile insect technique (pgSIT), which could generate flightless females and sterile males using a CRISPR-based approach ⁷” in Introduction part paragraph 2.

6. Line 68, investigation of their reproduction, what does “their reproduction” mean here?

Answer:

Thanks for your question. We have corrected as “a thorough investigation of mosquito reproduction” in in Introduction part paragraph 2.

7. Line 215, it is mainly expressed in sperm. It should be testis, rather than sperm. Otherwise, the expression should also be observed in the spermatheca.

Answer:

Thanks for your suggestions. We have corrected as “it is mainly expressed in testis” in Result part section “LAPI proteins localize to *Ae. aegypti* sperm”. Due to the excessive expression level of *LAPI* in the testes, its RNA levels in the spermatheca could not be shown well in our previous figure. In the revision, we have divided the Y-axis into two parts, and then the expression of *LAPI* in different tissues can be clearly observed (Fig.3a).

Fig. 3a mRNA abundance of *LAP1* in different mosquito tissue at the G2 (n=3). Data were normalized to the expression level of ovary ♀.

8. Line 419-421, there is no male flight ability test in the supplemental data. Instead, the sound-seeking assay is included, which is very unclear why this assay is performed, what the results mean, and how significant these results are related to the population suppression in the field as claimed in the discussion.

Answer:

Thank you for pointing this out. The supplementary figure legends in the paper are labeled as S1, S2, S3, S4, and S5, which may not be displayed correctly due to the names. We have corrected as “Supplementary Fig. 1, Supplementary Fig. 2, Supplementary Fig. 3, Supplementary Fig. 4 Supplementary Fig. 5” in the Supplementary information file.

Previous studies showed that the buzz of a flying female mosquito acts as a mating signal, attracting males (Cator LJ, Arthur BJ, Harrington LC, Hoy RR. Harmonic convergence in the love songs of the dengue vector mosquito. *Science*. 2009 Feb 20;323(5917):1077-9. doi: 10.1126/science.1166541. Epub 2009 Jan 8. PMID: 19131593; PMCID: PMC2847473.). Typically, the behaviorally salient frequency component of flight tone is the fundamental frequency of wing beat, between 300-600

Hz depending on species (Clements A N. The biology of mosquitoes. Volume 2: sensory reception and behavior[M]. CABI publishing, 1999.). Generally, 400-Hz tone was used to attract male *Ae. aegypti*. Hence, the ability of sterilized, or genetically modified males to modulate their flight tones could be a useful behavioral bioassay for the sterilization program. To investigate whether the LAP1 mutation affects the ability of male mosquitoes to seek females, sound-seeking assay were performed.

Supplementary Fig. 5a showed the speed of WT (left panel) and *LAP1*^{-/-} males (middle panel). The right panel was the variation in speed between WT♂ and *LAP1*^{-/-}♂ ($P=0.54$). Supplementary Fig. 5b displays the position frequency and prefer index to sound. In other words, this figure demonstrated the ability of male mosquitoes to seek females. Hence, sound-seeking was used as an indicator of a male reproductive fitness in this study.

9. There is no legend for the supplemental figures.

Answer:

Thank you for pointing this out. The supplementary figure legends in the paper are labeled S1, S2, S3, S4, and S5, which may not be displayed correctly due to the names. We have corrected as “Supplementary Fig. 1, Supplementary Fig. 2, Supplementary Fig. 3, Supplementary Fig. 4 Supplementary Fig. 5” in the Supplementary information file.

Reviewer #2 (Remarks to the Author):

Summary and Recommendation

In this study Sun and colleagues investigate the role of protein LAP1 in *Aedes aegypti* reproduction. Females with reduced or ablated LAP1 proteins produced through RNAi and CRISPR, lay significantly less eggs and those eggs are less likely to hatch. They also uncover an apparent role of LAP1 in male physiology. Not only do they find that gene encoding LAP1 is highly expressed in sperm, they also use crosses with LAP1 deficient CRISPR lines to show that females mating with males lacking in LAP1 are less fecund and fertile. Interestingly, this effect on fertility is age dependent. Matings with older LAP1 deficient males

result in smaller proportions of eggs that hatch from matings. Finally, the study presents a set of experiments designed to test the competitive ability and safety of a LAP1 knockout line.

The data presented are exciting and the SWATH-MS, and physiological aspects in particular are very much worthy of publication. However, there are critical details about the methodology and analysis missing from the manuscript that prevents us from recommending the manuscript for publication in its current form. We have additional concerns about the interpretation of some of the findings. Overall, the study reveals exciting aspects of basic mosquito physiology and reproductive biology. We would recommend focusing on these aspects instead of extrapolating into the applicability of these findings for mass releases. We found these to be unconvincing and they do a disservice to the rest. We have detailed these major concerns below and provided a list of minor concerns which the authors may wish to address.

Major Concerns:

1. There are no methods provided (at least in what we were able to find as reviewers) for the experiments reported in Figure 1 A, C, D.

Answer:

Thank you for pointing this out. We added in the Statistical analysis of revised manuscript such as below.

“Proteomic data were performed using <http://www.bioinformatics.com.cn>, which is used for analysis and visualization (Fig. 1a, 1b, 1c, Supplementary Fig. 1b, and 1c). Volcano plot analysis in Fig. 1a, GO enrichment analysis in Supplementary Fig. 1b, KEGG functional classification in Supplementary Fig. 1c, and the survival rate in Supplementary Fig. 5c were implemented by enhancedvolcano R package (1.13.2), Goplot R package (1.0.2), circlize R package (0.4.15), and survminer R package (0.4.9) in the R environment, respectively. Venn diagram in Fig. 1b and Sankey_dot plot in Fig. 1c were implemented by the matplotlib python package (3.7.0).” in Methods part section “Statistical analysis”.

2. The sample sizes for mating experiments as described in the methods and results do not match the figures. For example, for the initial crosses of homozygous and heterozygous LAP1 knockouts:

The methods indicate that 30 females were held with some unknown number of males for 3 days. After which individual females were separated into tubes and allowed to oviposit. It was from these females that the fecundity and fertility data was measured. Three independent assays were performed per condition. From this we would expect that for each treatment we would have data for ~90 females total. (Lines 491-501)

Answer:

We added more details about our experimental methods. We repeated each experiment three times independently, except newly added experiment of transmissibility of virus in *LAPI*^{-/-}, *LAPI*^{-/+}, and WT mosquitoes (supplementary Fig. 5d), and the figure in the article shows the results of an independent experiment. We added more details in the legends and methods such as below:

“Females that died during laying eggs were removed from the study. Three days later, the numbers of eggs were counted and separated into intact (unhatched) or broken (hatched). Each dot represents one mosquito. Data are shown as mean ± SEM. The experiments were repeated three times with similar results. Statistical significance of the egg deposition and hatchability was determined using a one-way ANOVA with multiple comparisons. Fisher's exact test was used to statistically analyze the ratio of females producing hatchable eggs to total mosquitoes. * $P < 0.05$, ** $P < 0.01$, *** $P < 0.001$. ns: not significant.” in Figure legend part section “Fig. 2 *LAPI* mutation in males limits female fecundity and fertility”.

“Females that died during laying eggs were removed from the study. Three days after oviposition, the numbers of eggs were counted and separated into intact (unhatched) or broken (hatched). The top of each column in **b** and **e** shows the number of hatched mosquitoes relative to total mosquitoes. Each dot represents one mosquito.

Data are shown as mean \pm SEM. Similar results were obtained in three independent experiments. The two-way ANOVA with multiple comparisons was used for the statistical analysis. *** $P < 0.001$.” in Figure legend part section “Fig. 5 Mating with *LAPI*^{-/-} males suppress female fertility”.

“Females that died during laying eggs were removed from the study. Three days later, the numbers of eggs were counted and separated into intact (unhatched) or broken (hatched). The top of each column in **b** and **e** shows the number of hatched mosquitoes relative to total mosquitoes. Each dot represents one mosquito. Data are shown as mean \pm SEM. Similar results were obtained in three independent experiments. Statistical significance of the egg deposition and hatchability was determined using a one-way ANOVA with multiple comparisons. * $P < 0.05$, ** $P < 0.01$, *** $P < 0.001$.” in Figure legend part section “Fig. 6 *LAPI*^{-/-} males compete with WT males to limit female fertility”.

SWATH-MS method: “Each group had three replicates.” in Methods part section “SWATH-MS”.

3. The written results indicate that there was a significant reduction in hatchability with no statistical information and refers the reader to Figure 2b (Lines 184-188).

Answer:

We corrected this as “The results revealed that knocking out *LAPI* resulted in a higher inhibition of mosquito hatchability (decreased by 93% and 100% after the first and second blood meals, respectively) compared to its effect on egg-laying (decreased by 31% and 41% after the first and second blood meals, respectively), suggesting that the effect of *LAPI* on embryonic development may be greater than that on ovarian development in mosquitoes (Fig. 2b).” in Result part section “CRISPR/Cas9-mediated deletion of *LAPI* affects the fecundity and fertility of female mosquitoes”.

4. Figure 2b The sample sizes appear for each treatment range from 18-36. Where did the rest of the females go? It appears as though a significant proportion (~ $\frac{2}{3}$) have died or not during the course of the study. This all needs to be explained clearly.

This is true of all of the mating assays. The reader must be able to determine how the reported sample size in the results have come from the methods as described.

In the Data availability section, it indicates that all the data underlying these figures is in a Source Data file. We were not able to access this as a reviewer to confirm sample sizes.

Answer:

We added more details in the legends and methods such as question 2. We will submit the Source Data file with the revised manuscript.

5. Clarification and justification of statistical analysis:

It is unclear what statistical tests have been applied to the data. It appears that Mann-Whitney tests were used on egg deposition and hatchability data (Lines 646-647). On Line 960 it indicates that a Fisher's exact test was used for hatchability data in some experiments, but not in others.

Answer:

Statistical analysis of hatchability was performed using a one-way ANOVA with multiple comparisons. Fisher's exact test was used to statistically analyze the ratio of females producing hatchable eggs to total mosquitoes, as in Fig. 2b and 2c (right panel) in Figure legend part section "*LAPI* mutation in males limits female fecundity and fertility".

6. It looks as though a series of pairwise comparisons have been made between the control in a given experiment and each experimental treatment. This type of pairwise comparison without appropriate

correction can inflate significance. Further, this approach does not account for the fact that individual females measures are nested within the replicate cage.

The authors need to clarify the statistical analysis for each experiment. Normally, we would recommend using a generalized linear mixed model or similar analysis to account for nested experimental design.

Answer:

We added more details in the legends such as below:

“Statistical significance of the egg deposition and hatchability was determined using a one-way ANOVA with multiple comparisons. Fisher's exact test was used to statistically analyze the ratio of females producing hatchable eggs to total mosquitoes.” in Figure legend part section “Fig. 2 *LAPI* mutation in males limits female fecundity and fertility”.

“The two-sided Mann-Whitney test was used for the statistical analysis.” in Figure legend part section “Fig. 3 The localization of *LAPI* in *Ae. aegypti*”.

“Statistical significance of the egg deposition and hatchability was determined using a one-way ANOVA with multiple comparisons” in Figure legend part section “Fig. 6 *LAPI*^{-/-} males compete with WT males to limit female fertility”.

7. Explanation and rationale for sound seeking assay:

The sound seeking assays (Lines 622-644) lack clear rationale and description. The dimensions of this box likely inhibit free flight (4 cm). The definition of metrics such as ‘large jump’ and ‘trajectory birth’ are missing along with an explanation of what these metrics indicate about male behavior. Please clarify how this relates to male mating behaviour. We were unable to find any statistical analysis of the flight assay.

Answer:

Thank you for pointing this out. The buzz of a flying female mosquito acts as a mating signal, attracting males (Cator LJ, Arthur BJ, Harrington LC, Hoy RR. Harmonic convergence in the love songs of the dengue vector mosquito. *Science*. 2009 Feb 20;323(5917):1077-9. doi: 10.1126/science.1166541. Epub 2009 Jan 8. PMID: 19131593; PMCID: PMC2847473.). Typically, the behaviorally salient frequency component of flight tone is the fundamental frequency of wing beat, between 300-600 Hz depending on species (Clements A N. *The biology of mosquitoes. Volume 2: sensory reception and behaviour*[M]. CABI publishing, 1999.). Generally, 400-Hz tone was used to attract male *Ae. aegypti*. Hence, the ability of sterilized, or genetically modified males to modulate their flight tones could be a useful behavioral bioassay for the sterilization program. To investigate whether the LAP1 mutation affects the ability of males to seek females, sound-seeking assay were performed.

In the absence of any disturbance, a number of males spread evenly around the box at a constant rate. After being stimulated by the sound of females, males would sprint towards the sound source for a moment and continue to gather around it. We believe that males that are more sensitive to female sounds will have more mating opportunities. Thus, the change in speed after activation by the sound and the preference for the sound are used to reflect the mating ability of the males in the experiments.

We believe that the size of the box does not interfere with free flight. We have the following reasons. Firstly, the mosquitoes had a steady speed inside the box without stimulation, which indicates that the mosquitoes flew smoothly and stayed healthy. Secondly, the mosquitoes in the experiments got used to the box for three days and rarely hit the wall. Thirdly, the phenomenon of each experiment was obvious and stable. Finally, this behavioral paradigm can also measure different responses to different sounds in other unpublished experiments, which shows the repeatability and feasibility.

We used ctrax, an open-source multi-target tracking software, to measure the speed and location of each mosquito. In the process of mosquito location tracking, there may be some identification errors that affect the accuracy of the computational results. Some errors could result in a large spatial span for a particular mosquito, which was called a "large jump". Some errors resulted from a temporary pause in the mosquito's tracking,

causing the same mosquito to be identified as a new individual, which was called "trajectory birth". And some mosquitoes switch identities due to staggered flight paths. After tracking, we will manually correct these errors.

The *P*-value was determined by a two-sided unpaired *t*-test.

8. Details and interpretation of viral infection data

Critical experimental details missing here including number of females fed and temperature at which females were held. Viral loads are measured at a single time point which may not be representative of infection dynamics (Line 640).

No difference in infection load at a single temperature and time point is not sufficient evidence to support the statement on Line 340-344 about safety. At best these experiments are preliminary and either should be expanded or removed from the manuscript.

Answer:

We corrected as suggested. We added more details in the paper, legends and method such as below:

“Importantly, *LAPI* deletion did not alter the ability of *Ae. aegypti* to transmit ZIKV or DENV in comparison with WT mosquitoes, suggesting the relative biosafety of the *LAPI*^{-/-} mutants (Supplementary Fig. 5d)” in Result part section “Fitness and safety assessments of *LAPI*^{-/-} males”.

Supplementary Fig. 5d Transmissibility of virus in *LAPI*^{-/-} mosquitoes. Seventy female mosquitoes in *LAPI*^{-/+} ♂ × *LAPI*^{-/+} ♀, *LAPI*^{-/+} ♂ × *LAPI*^{-/+} ♀, and WT ♂ × WT ♀ groups were infected with ZIKV or DENV. Total RNA was extracted from a single mosquito at 3-, 7-, and 10-days post-infection, and the viral RNA level was measured by qPCR. The top of each column shows the number of infected mosquitoes relative to total mosquitoes. Each dot represents one mosquito. Data are shown as mean ± SEM. The experiments were repeated twice with similar results. Statistical significance was determined using a one-way ANOVA with multiple comparisons. ns: not significant.

“Seventy females from the *LAPI*^{-/+} ♂ × *LAPI*^{-/+} ♀, *LAPI*^{-/+} ♂ × *LAPI*^{-/+} ♀, and WT ♂ × WT ♀ groups were mated with males at a 1:1 ratio. Three- to four-day-old WT, *LAPI*^{-/+} and *LAPI*^{-/-} female mosquitoes were starved for 15-18 h before viral infection. Mouse blood containing 1.0×10^6 FFU/mL ZIKV or DENV was preheated for 30 min at 37 °C. The mixture was then fed to females for 30 min at 37 °C through an *in vitro* membrane feeding apparatus. Mosquitoes were anesthetized for 15 min at 4 °C, and females with a full-blood meal were transferred to new containers. The mosquitoes were then reared in an incubator at 28°C, 80% humidity, and a 12/12-hour light-dark cycle. Total RNA was extracted from a single mosquito at 3-, 7-, and 10-days post-infection, and the viral RNA level was measured by qPCR.” in Methods part section “Viral infection”.

9. Interpretation of competitive mating assays.

Our understanding of the data presented in Figure 5 is that *LAP1*^{-/-} males allowed to mate with WT females in the absence of competition result in extremely low hatchability. The curve suggests that a female mating with a 3.8398-day old *LAP1*^{-/-} male would be predicted to have a 10% hatch rate and that this rate would continue to fall as female mates with older males. [NOTE: methods do not clarify how many males are held with the 30 females.]

Answer:

We added details about our experimental methods in Fig. 5. Due to the difference in hatchability and fertile female ratio between the first and second blood meals in the *LAP1*^{-/-}♂ × WT♀, we aimed to understand whether blood feeding for the indicated days (0, 4, 8, 12, and 16 days) after mating influenced hatching rate (Fig. 2c). In general, the females are not capable of blood-feeding until three days after hatching. The results showed that as the number of days the female mosquitoes were blood-feeding increased, the hatchability continued to decrease (Fig. 5b). Meanwhile, Fried's Competitiveness Index was used to quantify the mating competitiveness of *LAP1*^{-/-} males in blood-feeding experiments on females for indicated days (0, 4, 8, 12, and 16 days after mating), and it was 0.95, 0.98, 1.09, 1.05, and 1.07, respectively, indicating that the mating competitiveness of *LAP1*^{-/-}♂ and WT♂ was comparable. A value of $c = 1$ corresponds to compatible and wildtype males having equal mating competitiveness; $c < 1$ to compatible males being less competitive for mates; and $c > 1$ meaning that compatible males are more competitive than wildtypes. The curve showed that in the absence of competition, females were fed blood at 3.84 days after mating with *LAP1*^{-/-} males, resulting in a decrease in overall hatching rate to 10% (Fig.5c).

Fig. 5e showed that WT females have progressively reduced hatchability when mated with the indicated day-old *LAP1*^{-/-} males. Importantly, we also measured the mating competitiveness of *LAP1*^{-/-} males under the same conditions using Fried's Competitiveness Index, which was 1.09, 1.83, 1.973, 2, 1.68, and 1.87 for mating with

at 1-, 3-, 5-, 7-, 9-, and 11-day-old males, respectively, indicating that $LAPI^{-/-}\sigma$ were comparable to or more competitive than $WT\sigma$ for mating. Older $LAPI^{-/-}$ males not only induced better sterility, but are also competitive enough. Curve fitting demonstrated that in the absence of competition, WT females mated with 6.2-day-old $LAPI^{-/-}$ males were able to reduce the overall hatching rate to 10% (Fig. 5f).

Further details are provided in the Figure legends below:

“Fig. 5 Mating with $LAPI^{-/-}$ males suppresses female fertility. a Schematic diagram of the stored sperm fertility assay. After mating with 30 1-day-old WT or $LAPI^{-/-}$ males for 3 days, 30 WT females were blood fed at indicated days (0, 4, 8, 12, and 16 days) post mating. And egg hatchability was measured. **b** The fertility of 30 WT females following blood feeding at indicated days after mating with 30 1-day-old WT or $LAPI^{-/-}$ males (n=30). **c** Curve fitting of the same data from **b**. The dashed line shows the point, at which mosquitoes were blood fed after mating with $LAPI^{-/-}$ males resulted in a 10% of female fertility. The red curve was served as the fitted curve. **d** Schematic diagram of the sperm fertility assay. Thirty WT females were mated with 30 indicated day-old (1-, 3-, 5-, 7-, 9-, 11-day-old) WT or $LAPI^{-/-}$ males. And blood meal was given after mating for 3 days. Egg deposition and hatchability were measured later. **e** The fertility of 30 WT females mated with 30 indicated old-WT or $LAPI^{-/-}$ males (n=30). **f** Curve fitting of the same data from **e**. The dashed line shows the point at which mating with indicated old- $LAPI^{-/-}$ males resulted in a 10% reduction in female fertility. The red curve was represented the fitted curve. Females that died during laying eggs were removed from the study. Three days after oviposition, the numbers of eggs were counted and separated into intact (unhatched) or broken (hatched). The top of each column in **b** and **e** shows the number of hatched mosquitoes relative to total mosquitoes. Each dot represents one mosquito. Data are shown as mean \pm SEM. Similar results were obtained in three independent experiments. The two-way ANOVA with multiple comparisons was used for the statistical analysis. *** $P < 0.001$.”

Further details are provided in the methods below:

“Fried's competitiveness index was used to calculate the mating competitiveness

of $LAPI^{-/-}$ males in blood-feeding experiments on females for indicated days and mating experiments between females and indicated age mutants. In the competitiveness index developed by Fried, it assumes that the proportion of hatching eggs (H_m) is determined from a mixture of S (incompatible) and W (WT) insects as follows:

$$H_m = \frac{WH_w}{W + S} + \frac{SH_s}{W + S}$$

H_w and H_s represent the expected proportions of eggs hatching, when WT females are exposed to WT males or incompatible males, respectively. The underlying equation assumes the equal mating competitiveness between WT and incompatible males. However, it is more reasonable to assume unequal competitiveness, thus leading to a revised equation as follows:

$$H_m = \frac{WH_w}{W + cS} + \frac{cSH_s}{W + cS}$$

C represents the coefficient of mating competitiveness. The parameter c can be regarded as the probability, $\frac{P_s}{\{1-P_s\}}$, which means a female will mate with an incompatible male rather than a WT male, and both are in equal numbers (i.e., $W = S$). And P_s represents the probability that a mated female will choose to mate with an incompatible male rather than a WT male. The above equation could be rearranged to acquire to yield c . Thus, we exposed WT females to WT males, $LAPI^{-/-}$ males, and 1:1 mixtures of WT and $LAPI^{-/-}$ males to acquire $H_{WT,i}$, $H_{LAP1,i}$, and $H_{m,i}$, which are the hatching rates in condition i , and calculated c_i in condition i using the equation

$$c = \frac{W(H_w - H_m)}{S(H_m - H_s)}$$

Thus, we exposed WT females to WT males, $LAPI^{-/-}$ males, and 1:1 mixtures of WT and $LAPI^{-/-}$ males to obtain $H_{WT,i}$, $H_{LAP1,i}$, and $H_{m,i}$, the proportions of eggs that hatched in condition i , and calculated c_i in condition i using the equation

$$c_i = \frac{H_{WT,i} - H_{m,i}}{H_{m,i} - H_{LAP1,i}}$$

” in Methods part section “Fried's Competitiveness Index for suppressing female fertility in $LAPI^{-/-}$ males”.

10. In Figure 6, the reported hatch rate when 29 $LAP1^{-/-}$ males

compete with 1 WT male is still >20%. This suggests that the WT male is mating a disproportionate number of females and the *LAP1* ^{-/-} males are highly uncompetitive. This may be an issue with the details of these assays not being clear, but certainly needs clarification. [NOTE: Figure 6 c and f label the x axes with 'days' rather than presumably 'ratio']

Answer:

Thank you for pointing this out. We changed "days" to "ratio" in Fig. 6c and 6f. As shown in Fig. 2c and the Source Data file, hatchability was relatively variable among individuals, ranging from 0-56.6% in the *LAP1*^{-/-}♂ × WT♀ group after the first blood meal, compared to more than 80% in the WT♂ × WT♀ group. Correspondingly, only two females were found with hatchability greater than 60% (67.6% and 81.6%), indicating that WT males were not mating with too many females (Fig. 6b and the Source Data file). To further confirm the ability of *LAP1*^{-/-} males to compete with WT males for mating, we mixed 30 virgin WT females with 6-day-old *LAP1*^{-/-}♂ to 1-day-old WT♂ in a 29:1 ratio for 3 days. Hatchability was then assessed after the first and second blood meals. The addition of 29:1 resulted in a significant additional reduction in female hatchability (reduced to an average of 22.6% and 7.3% after the first and second blood, respectively) (Figure below). Importantly, only two females showed a hatching rate above 60% after the first blood meal, while the remaining females were further suppressed after the second blood meal.

Figure. The fertility of WT females mated with $LAP1^{-/-}$ and WT males at 29:1 ratio after the first and second blood meals. 6-day-old $LAP1^{-/-}$ and 1-day-old WT males were introduced simultaneously at 29:1 ratio (30 males in total) to mate with 30 WT females until mosquitoes laid eggs (n=30). Females that died during laying eggs were removed from the study. Three days later, the numbers of eggs were counted and separated into intact (unhatched) or broken (hatched). The top of each column shows the number of hatched mosquitoes relative to total mosquitoes. Each dot represents one mosquito. Data are shown as mean \pm SEM. Statistical significance of the egg deposition and hatchability was determined using a one-way ANOVA with multiple comparisons. * $P < 0.05$, ** $P < 0.01$, *** $P < 0.001$.

11. Further in these competitive assays, the ratio of LAP1 $-/-$ males to WT males is varied, but the intensity of competition for females overall remains constant because the total number of males to females is equal across the assays. This ration does not represent the operational sex ratio in the field. This point needs to be addressed.

Answer:

We examined 50 virgin WT females mated with 6-day-old $LAP1^{-/-}$ ♂ and 1-day-old WT ♂ at various ratios for 3 days to assess hatchability after the first and second blood meals. The ratios of $LAP1^{-/-}$ ♂ to WT ♂ were 0:1, 1:1, 3:1, 5:1, 9:1, 14:1, and 29:1 for a total of 30 males. The results showed that the addition of 1:1, 3:1, 5:1, 9:1, 14:1 and 29:1 resulted in a lower hatchability after the first blood meal (63.1%, 53.1%, 49.5%, 46.7%, 36.6%, and 25.1%, respectively; hatchability at 0:1 ratio was 93.8% on average), which was similar to the results of mating with 30 female mosquitoes (64.7%, 52.6%, 49.2%, 44.3%, 35.5%, and 24.5%, respectively; hatchability at 0:1 ratio was 91.5% on average) (Figure below). In addition, hatchability was further suppressed after the second blood meal because the effect of LAP1 on the fertility of sperm stored by females became stronger over time.

Figure. The fertility of WT females mated with *LAPI*^{-/-} and WT males at 29:1 ratio after the first and second blood meals. 6-day-old *LAPI*^{-/-} and 1-day-old WT males were introduced simultaneously at different ratios (30 males in total) to mate with 50 WT females until mosquitoes laid eggs. Three days later, the numbers of eggs were counted and separated into intact (unhatched) or broken (hatched). The top of each column shows the number of hatched mosquitoes relative to total mosquitoes. Each dot represents one mosquito. Data are shown as mean ± SEM. The *P*-value was determined by one-way ANOVA with multiple comparisons and compared with the group of 0:1 group. * *P* < 0.05, ** *P* < 0.01, *** *P* < 0.001.

A single mating enables a female mosquito to store enough sperm in the spermathecae to contribute to fertilizing all the eggs during the lifetime. However, in this study, we only preliminarily evaluated the fitness of the *LAPI*^{-/-} males under laboratory conditions, such as equal total numbers of females and males, and excluded the effect of LAP1 on the hatching rate of females after second or multiple blood meals. Our subsequent work requires further reduction of residual fertility in *LAPI*^{-/-} males and the establishment of a comprehensive release model to specifically assess the release rate.

Minor Concerns:

1. Author ASR is not mentioned in the Author Contributions. The contribution of this author to the work needs to be specified.

Answer:

Thanks, we have corrected as “X.S, X.W, K.S, X.L, J.S, ASR and Z.Z contributed to reagents/materials/analysis tools.

2. It is unclear why the authors choose to focus exclusively on male physiology. The justification provided is that the phenotype is more pronounced in the matings between *LAP1*^{-/-} males and WT females, but there is still a significant difference in hatchability and egg deposition in the first gonotrophic cycle in the matings involving WT males and *LAP1*^{-/-} females suggesting an important role in female reproduction as well. Some extended discussion of this interesting result is warranted.

Answer:

We agree with you that the presence of *LAP1* is essential for ovarian and egg development. As is shown in Fig. 3c, the WT mosquitoes almost completed blood digestion at 48 h PBM, whereas the midguts of *LAP1*^{-/-} mosquitoes were still full with blood, which might account for low egg deposition. However, in this paper, we were not able to reveal the mechanism of the effect of *LAP1* on both males and females simultaneously. We will continue working to understand how *LAP1* affects the reproduction of female mosquitoes in the further.

3. Why were females not provided a bloodmeal on the same day as their matings? Wouldn't this be the most natural scenario? The earliest females receive these in Figure 5 and 6 experiments is 3 days post mating.

Answer:

This is because it is necessary to ensure that all females are fully mated. The physiological characteristics of mosquitoes are that the males and females do not mate on the first day of hatching, and the females are not capable of blood-feeding well until three-four days after hatching. Therefore, our experiment was to feed the females three days after mating.

4. Figure 2 - For Hatching rate (middle panels Part B and C) the y axis should stop at 100%. The use of different colours for each bloodmeal are unnecessary. We suggest single colors would be clearer. There is small typo in the y-axis for the right panels in BC should read hatched/total mosquito

Answer:

For the hatching rate (Figure 2 middle panel parts B and C), stopping the Y-axis at 100% would destroy the shape of the violin graph, so it ranges from 0-120%. In middle panels Part B and C, we corrected the colors as suggested. And we changed to “mosquito” in the y-axis.

5. Figure 3- For Part A please display the individual biological replicates (as in Part C). For Part D there is not a clear description where exactly this data have come from or the meaning of the y-axis.

Answer:

We corrected as suggested in Fig. 3a. We add more details in Figure legend part “Fig. 3 The localization of *LAP1* in *Ae. aegypti*.” such as below:

“Temporal expression of *LAP1* during different stages of embryonic development in $WT_{\text{♀}} \times LAP1^{-/-}_{\text{♂}}$ and $WT_{\text{♂}} \times WT_{\text{♀}}$ groups (n=3). Total RNA was extracted from eggs collected from the pairs $LAP1^{-/-}_{\text{♂}} \times WT_{\text{♀}}$ and $WT_{\text{♂}} \times WT_{\text{♀}}$ at different periods (0-1, 1-2, 2-4, 4-6, 6-8, 8-10, 10-12, and 20-22 h) after the first and second blood meals. RNA level of *LAP1* was measured using qPCR. Data were normalized to the expression level of $WT_{\text{♂}} \times WT_{\text{♀}}$ at 0-1 h post first blood meal”.

6. Figure 5- the legend discusses sperm motility but that was not measured here. We believe fertility may be a more appropriate term, or simply using hatchability.

Answer:

We have corrected “motility” to “fertility” as suggested.

7. Figure 6- Typo in x-axis of parts C and F- remove "days", Here again, difficult to judge due to lack of detail, but the error bars do not appear to vary with the spread of the data. This could just be visual thinking, but would be good to confirm these are accurate?

Line 470 in SWATH-MS methods: the acronym DIA is not explained.

Provide references for all software and programmes- e.g. Peakview

We changed "days" to "ratio" in Fig. 6c and 6f.

We checked the source data and found no errors in the graphs. Data are presented as mean \pm SEM in Fig. 6b and 6e. We will submit the Source Data file at this time.

The full name of DIA is explained in Result part section "Analysis of spermathecal fluid proteome by SWATH-MS in *Ae. aegypti*".

We add more details in the methods such as below:

"Briefly, the protein is first extracted, enzymatically digested and purified. The enzymatically digested peptides are then subjected to a data-dependent acquisition (DDA) mode to obtain the total ion flow map of the mass spectrometry signal. Peptides were identified using a Triple time-of-flight (TOF) 6600 instrument interfaced with an Agilent 1100 HPLC system (equipped with the ChromXP CL-3 μm , 120 \AA , 350 $\mu\text{m} \times 0.5 \text{ mm}$, 1.8 μm 150 $\mu\text{m} \times 20 \text{ cm}$). To acquire protein identification data, the results got from DDA were retrieved by MaxQuant⁷². In the next step, data were incorporated into Peakview to establish an ion library without the need to import shared peptides⁷³. Samples from 500 individual females at the virgin, 72 h PE and 5 days PBM stages were examined using DIA mass spectrometry. The detected spectra of the DIA samples were matched, and quantitative information were extracted from the ion library using Peakview. Finally, the protein identification results from Peakview were statistically analyzed." In Methods part section "SWATH-MS".

Reviewers' Comments:

Reviewer #1:

Remarks to the Author:

All of my previous comments have been addressed. Thus, I support the publication of this manuscript.

Reviewer #2:

Remarks to the Author:

Many elements of this manuscript are very exciting. The authors have clarified and addressed several of the points in the original review. However, there are still outstanding issues that remain.

Principle among these is the lack of clear presentation of the methods and statistical analyses. Ideally, the reader should be able to repeat the experiment and analysis for each result presented in the figures from the methods section. As the manuscript is written now, this is not possible without pulling details from the existing methods, results, and figure legends. We have included some specific comments about this below, but in general this aspect is very difficult to understand. For each experiment, the numbers of males and females in each cage, the exact measures taken and the statistical tests applied need to be specified. For all statistical tests, assumptions for using this test were satisfied and exactly which variables are included as predictors are needed.

Second, there is not satisfactory support for the statement on MS Line 420-423: "In our study, we used the CRISPR/Cas9 system to generate low-fertility LAP1-/- males that show no differences from WT males in terms of physiological parameters, including longevity, flight ability, and virus transmission capacity, suggesting the LAP1 mutants are as safe and robust as WT." The study contains a measure of longevity under laboratory conditions which may not reflect field survival. The flight assay does not test flight ability or capacity. At best, it is a measure of phonotactic responses in these males. Viral replication did not differ for one strain of DENV measured at one temperature in one genetic background. While we agree that the results are in a positive direction, this is over extrapolating their meaning without any discussion of caveats.

We were not satisfied by the response to our question about the flight assay. Cator et al. 2009, describes a phenomena known as harmonic convergence, which is not assessed in this study. The attraction of males to tones in the range of 300-600Hz (phonotaxis) is relevant for male mating success and is measured here. The assay described measured the frequency with which males were found near the sound emitting speaker and the control speaker. It is not clear how the preference for the sound producing speaker and speed at which males respond to that sound are a proxy for male flight ability.

Specific Comments to Address:

Figure 1- Which analysis has produced the P-values and fold changes used to create the volcano plots in part a and c? Identifying the package does not explain the analysis or statistical tests applied.

The experimental methods for Figure 2 are still unclear in the methods. We suggest creating a section for this figure in the methods section as has been done for Fig 5 and 6. Also for the analysis in Figure 2, it is indicated that a one-way ANOVA has been used. Based on what has been written here, we assume this is a ANOVA used to compare among the different crosses as. How were replicate effects taken into account? Which post-hoc pairwise comparison method was used (for example, Tukey's?) and how was this corrected for multiple comparisons?

We have similar questions about the analysis presented in Fig 5 and 6. How was replicate taken into account statistically and in Figure 5 how were interactions between cross type and male age or cross type and days between mating and bloodfeeding taken into account? Methods for post-hoc and corrections for multiple comparisons are required. Also, many of these outcome variables do not appear to be normally distributed which would make an ANOVA an inappropriate test. Please confirm

data fit model assumptions.

The information in the statistical analysis section (MS Line 702-705) is so vague that it is not informative. We suggest describing each statistical approach with the corresponding section of experimental methods.

There are several places throughout the MS where the term "significant difference" is used without statistical support. Instead, please use a different term here such as "large" or "notable". Significance should only be used in tandem with statistical analysis.

Thank you for including clarification of the sample sizes. To keep this clear, please give the planned sample size in the methods for each experiment (30 or 50) and the realized sample size in the figure legend (as you have done) with your clarification that many females (%) are lost in all assays.

In the 'safety and fitness assessments of LAP1^{-/-} males' experiments it isn't clear how the ratios of males of different genotypes was achieved; a ratio of 3:1 WT to LAP1^{-/-} males (MS line 317-319) isn't possible with 30 males in total.

REVIEWER COMMENTS

Reviewer #2 (Remarks to the Author):

Many elements of this manuscript are very exciting. The authors have clarified and addressed several of the points in the original review. However, there are still outstanding issues that remain.

Principle among these is the lack of clear presentation of the methods and statistical analyses. Ideally, the reader should be able to repeat the experiment and analysis for each result presented in the figures from the methods section. As the manuscript is written now, this is not possible without pulling details from the existing methods, results, and figure legends. We have included some specific comments about this below, but in general this aspect is very difficult to understand. For each experiment, the numbers of males and females in each cage, the exact measures taken and the statistical tests applied need to be specified. For all statistical tests, assumptions for using this test were satisfied and exactly which variables are included as predictors are needed.

Answer:

Thank you for pointing this out. We have used more appropriate statistical tests, described the number of females and males in each experiment, and added details within the legends, as shown below:

Fig. 2 *LAPI* mutation in males limits female fecundity and fertility. **a** Analysis of female fecundity and fertility after RNAi knockdown of ten candidate genes ($n > 37$). dsRNA was microinjected into the thorax of 1-day-old virgin female mosquitoes (50 females per group). After a 3-day recovery period, females were mated with 50 males for 3 days before the blood meal. The numbers of eggs were counted at 6 days PBM, and subsequently these eggs were used to calculate the hatchability. Microinjection resulted in a proportion of mosquitoes dying or being unwilling to have blood meal; they were removed from the experiment. Mosquitoes injected with *EGFP* dsRNA were used as controls. The q -value was calculated by comparing the experimental group with

the *EGFP* dsRNA-treated group. **b-c** Comparison of the egg deposition among $WT^{\♂} \times WT^{\♀}$, $LAPI^{-/-\♂} \times LAPI^{-/-\♀}$, and $LAPI^{-/+ \♂} \times LAPI^{-/+ \♀}$ groups after the first (**b**; $n > 62$) and second (**c**; $n > 48$) blood meals (50 females were paired with 50 males per group). **d-g** Comparison of the hatchability among $WT^{\♂} \times WT^{\♀}$, $LAPI^{-/-\♂} \times LAPI^{-/-\♀}$, and $LAPI^{-/+ \♂} \times LAPI^{-/+ \♀}$ groups after the first (**d** and **f**; $n > 62$) and second (**e** and **g**; $n > 48$) blood meals (50 females were paired with 50 males per group). The percentages in **f** and **g** indicate the rate that mosquitoes produce hatchable eggs. **h-i** Comparison of the fecundity among $WT^{\♂} \times WT^{\♀}$, $LAPI^{-/-\♂} \times WT^{\♀}$, and $WT^{\♂} \times LAPI^{-/-\♀}$ groups after the first (**h**; $n > 62$) and second (**i**; $n > 54$) blood meals (50 females were paired with 50 males per group). **j-m** Comparison of the fertility among $WT^{\♂} \times WT^{\♀}$, $LAPI^{-/-\♂} \times WT^{\♀}$, and $WT^{\♂} \times LAPI^{-/-\♀}$ groups after the first (**j** and **l**; $n > 62$) and second (**k** and **m**; $n > 54$) blood meals (50 females were paired with 50 males per group). The percentages in **l** and **m** indicate the rate that mosquitoes produce hatchable eggs. The top of each column in **b**, **c**, **h** and **i** indicates the numbers of mosquitoes that laid eggs, while in **d**, **e**, **j** and **k** shows the numbers of hatched mosquitoes relative to total egg-laying mosquitoes. Females that died during egg laying were removed from the study. Each dot represents one mosquito. The experiments were independently performed twice with 50 females per experiment. A two-tailed Mann-Whitney test of variance was conducted to detect any significant variation between two replicates, and no significant differences were detected. Then results were combined for further analyses. Egg deposition and hatchability were analyzed using the Kruskal-Wallis test followed by Dunn's post-hoc tests with Benjamini-Hochberg adjustment for multiple comparisons (**a**, **b**, **c**, **d**, **e**, **h**, **i**, **j**, and **k**). And Fisher's exact test was used to statistically analyze the ratio of females producing hatchable eggs relative to total mosquitoes (**f**, **g**, **l**, and **m**). Data in **a**, **b**, **c**, **d**, **e**, **h**, **i**, **j**, and **k** are shown as mean \pm SEM. * $q < 0.05$, ** $q < 0.01$, *** $q < 0.001$. ns: not significant.

Fig. 3 The localization of LAP1 in *Ae. aegypti*. **a** mRNA abundance of *LAP1* in different mosquito tissue at the G2 ($n=3$). The experiment was replicated three times, and tissues collected from 20 individual females and males at 72 h PE were served as a

single replicate per group. Data were normalized to the expression level of ovary♀. Statistical significance was determined using the one-way ANOVA test followed by Tukey's post hoc tests with Benjamini-Hochberg adjustment for multiple comparisons. *** $q < 0.001$. **b** Images of sperm in WT, $LAPI^{-/+}$, and $LAPI^{-/-}$ male mosquitoes. Sperm was dissected, and LAP1 protein was detected by immunofluorescence assay using mouse anti-LAP1 antibody (green). Nucleus was stained by Hoechst 33258 (blue). Images were visualized under a confocal microscope. Scale bar: 20 μm . **c** The ovarian phenotypes (left panel) and follicle lengths (right panel) of WT and $LAPI^{-/-}$ female mosquitoes were detected at 48 h PBM ($n > 26$). Results are combined from three batches of mosquitoes, and each dot represents the length of a single follicle. Images were captured from CellSens software (version 1.6) using an Olympus SZX16 stereoscopic microscope at $5 \times$ magnification (scale bar: 500 μm). The follicle lengths were measured in CellSens software (version 1.6). The p -value was determined by two-tailed Mann-Whitney test. **d** Temporal expression of $LAPI$ during different stages of embryonic development in $LAPI^{-/-}\text{♂} \times \text{WT}\text{♀}$ and $\text{WT}\text{♂} \times \text{WT}\text{♀}$ groups ($n=3$). The experiment was repeated three times, and total RNA was extracted from eggs collected from the pairs $LAPI^{-/-}\text{♂} \times \text{WT}\text{♀}$ and $\text{WT}\text{♂} \times \text{WT}\text{♀}$ at different periods (0-1, 1-2, 2-4, 4-6, 6-8, 8-10, 10-12, and 20-22 h) after the first and second blood meals as individual replicates for each group. RNA level of $LAPI$ was measured using qPCR. Data were normalized to the expression level of $\text{WT}\text{♂} \times \text{WT}\text{♀}$ at 0-1 h post first blood meal. Statistical significance was determined using the multiple Mann-Whitney test with Benjamini-Hochberg adjustment. ns: not significant. Data in **a**, **c** and **d** are represented as mean \pm SEM. The experiments were repeated three times with similar results.

Fig. 5 Mating with $LAPI^{-/-}$ males suppresses female fertility. **a** Schematic diagram of the stored sperm fertility assay. After mating with 1-day-old WT or $LAPI^{-/-}$ males for 3 days, WT females were blood fed at indicated days (0, 4, 8, 12, and 16 days) post mating. And egg hatchability was measured. **b** The fertility of WT females following blood feeding at indicated days after mating with 1-day-old WT or $LAPI^{-/-}$ males (30 females were paired with 30 males per group) ($n > 55$). The q -value was

calculated by comparing the experimental group with the 0 days post mating group. **c** Curve fitting of the same data from **b**. The dashed line shows the point, at which mosquitoes were blood fed after mating with *LAPI*^{-/-} males resulted in a 10% of female fertility. The red curve was served as the fitted curve. **d** Schematic diagram of the sperm fertility assay. WT females were mated with indicated day-old (1-, 3-, 5-, 7-, 9-, 11-day-old) WT or *LAPI*^{-/-} males. And blood meal was given after mating for 3 days. Egg deposition and hatchability were measured later. **e** The fertility of WT females mated with indicated old-WT or *LAPI*^{-/-} males (30 females were paired with 30 males per group) (n>57). The *q*-value was calculated by comparing the experimental group with the 1-day-old group. **f** Curve fitting of the same data from **e**. The dashed line shows the point at which mating with indicated old-*LAPI*^{-/-} males resulted in a 10% reduction in female fertility. The red curve was represented the fitted curve. Females that died during laying eggs were removed from the study. Three days after oviposition, the numbers of eggs were counted and subsequently separated into intact (unhatched) or broken (hatched). The top of each column in **b** and **e** shows the number of hatched mosquitoes relative to total mosquitoes. Each dot represents one mosquito. The experiments were independently performed three times with 30 females per experiment. A Kruskal-Wallis test of variance was conducted to detect any significant variation among three replicates, and no significant differences were detected. Then results were combined for further analyses. Statistical significance was determined using the multiple Mann-Whitney test with Benjamini-Hochberg adjustment (**b** and **e**). Data are shown as mean \pm SEM. * *q* < 0.05, ** *q* < 0.01, *** *q* < 0.001.

Fig. 6 *LAPI*^{-/-} males compete with WT males to limit female fertility. **a** Scheme of the mating competition assay of 1-day-old male mosquitoes (30 females were paired with 30 males per group). The ratios of *LAPI*^{-/-}♂ to WT ♂ were 0:1 (0 *LAPI*^{-/-}♂ and 30 WT ♂), 1:1 (15 *LAPI*^{-/-}♂ and 15 WT ♂), 2:1 (20 *LAPI*^{-/-}♂ and 10 WT ♂), 5:1 (25 *LAPI*^{-/-}♂ and 5 WT ♂), 9:1 (27 *LAPI*^{-/-}♂ and 3 WT ♂), 14:1 (28 *LAPI*^{-/-}♂ and 2 WT ♂), and 29:1 (29 *LAPI*^{-/-}♂ and 1 WT ♂) for a total of 30 males in each group. **b** The fertility of WT females mated with 1-day-old *LAPI*^{-/-} and WT males at different ratios

was evaluated ($n > 56$). 1-day-old *LAPI*^{-/-} and WT males were introduced simultaneously at different ratios (30 males in total) to mate with 30 WT females, and females were given blood meal after 3 days. Egg hatchability was then measured. **c** Curve fitting of the same data from **b**. The dashed line shows the point at which mating with 1-day-old males of indicated *LAPI*^{-/-} to WT ratios resulted in a 50% reduction in female fertility. The fitted curve was marked with red. **d** Scheme of mating competition assay of 6-day-old male mosquitoes (30 females were paired with 30 males per group). **e** The fertility of WT females mated with *LAPI*^{-/-} and WT males at different ratios. 6-day-old *LAPI*^{-/-} and 1-day-old WT males were introduced simultaneously at different ratios (30 males in total) to mate with 30 WT females ($n > 58$). Blood meal was given after 3 days. And egg hatchability was measured. **f** Curve fitting of the same data from **e**. The dashed line shows the point at which mating with 6-day-old males of indicated *LAPI*^{-/-} to WT ratios resulted in a 50% reduction in female fertility. The fitted curve was marked with red. Females that died during laying eggs were removed from the study. The top of each column in **b** and **e** shows the number of hatched mosquitoes relative to total mosquitoes. Each dot represents one mosquito. The experiments were independently performed three times with 30 females per experiment. A Kruskal-Wallis test of variance was conducted to detect any significant variation among three replicates, and no significant differences were detected. Then results were combined for further analyses. Statistical significance of hatchability was determined using the Kruskal-Wallis test followed by Dunn's post-hoc tests with Benjamini-Hochberg adjustment for multiple comparisons (**b** and **e**). The q -value was calculated by comparing the experimental group with the 0:1 group. Data are shown as mean \pm SEM. * $q < 0.05$, ** $q < 0.01$, *** $q < 0.001$.

Second, there is not satisfactory support for the statement on MS Line 420-423: "In our study, we used the CRISPR/Cas9 system to generate low-fertility LAP1^{-/-} males that show no differences from WT males in terms of physiological parameters, including longevity, flight ability, and virus transmission capacity, suggesting the LAP1 mutants are as safe and robust as WT." The study contains

a measure of longevity under laboratory conditions which may not reflect field survival. The flight assay does not test flight ability or capacity. At best, it is a measure of phonotactic responses in these males. Viral replication did not differ for one strain of DENV measured at one temperature in one genetic background. While we agree that the results are in a positive direction, this is over extrapolating their meaning without any discussion of caveats.

Answer:

Thank you so much for the comment. We have added the following in the discussion:

In our study, we used the CRISPR/Cas9 system to generate low-fertility *LAPI*^{-/-} males that show no differences from WT males in terms of physiological parameters under laboratory conditions, including longevity and virus transmission capacity, suggesting that *LAPI* mutants may be as safe and robust as the WT, but further field experiments are needed to be proved under real field conditions.

We were not satisfied by the response to our question about the flight assay. Cator et al. 2009, describes a phenomenon known as harmonic convergence, which is not assessed in this study. The attraction of males to tones in the range of 300-600Hz (phonotaxis) is relevant for male mating success and is measured here. The assay described measured the frequency with which males were found near the sound emitting speaker and the control speaker. It is not clear how the preference for the sound producing speaker and speed at which males respond to that sound are a proxy for male flight ability.

Answer:

Unpublished studies from our lab were able to answer the question of the preference for the sound producing speaker and speed at which males respond to that sound are a proxy for male flight ability. To avoid impacting unpublished experiments, we decided to remove the sound-seeking assay.

Specific Comments to Address:

Figure 1- Which analysis has produced the P-values and fold changes used to create the volcano plots in part a and c? Identifying the package does not explain the analysis or statistical tests applied.

Answer:

Thank you for pointing this out. We added more details within the methods (SWATH-MS) and legends (Fig 1c), as indicated below:

Method of **SWATH-MS**: Spermathecae from 2000 individual female mosquitoes at the virgin, 72 h PE and 5 days PBM stages were pooled for the construction of a spectral library. In addition, spermathecae from 500 individual females at the same time point as above were collected for analysis. Each group had three replicates. Mass spectrometry was then performed by Beijing Protein Innovation Co., Ltd.

Briefly, the protein is first extracted, enzymatically digested and purified. The enzymatically digested peptides are then subjected to a data-dependent acquisition (DDA) mode to obtain the total ion flow map of the mass spectrometry signal. Peptides were identified using a Triple time-of-flight (TOF) 6600 instrument interfaced with an Agilent 1100 HPLC system (equipped with the ChromXP CL-3 μm , 120 \AA , 350 $\mu\text{m} \times 0.5 \text{ mm}$, 1.8 μm 150 $\mu\text{m} \times 20 \text{ cm}$). To acquire protein identification data, the results got from DDA were retrieved by MaxQuant⁶⁵. In the next step, data were incorporated into Peakview to establish an ion library without the need to import shared peptides⁶⁶. Samples from 500 individual females at the virgin, 72 h PE and 5 days PBM stages were examined using data-dependent acquisition (DIA) mass spectrometry. The detected spectra of the DIA samples were matched, and quantitative information were extracted from the ion library using Peakview. Finally, the protein identification results from Peakview were statistically analyzed. Fold change is the ratio between the mean quantitation values of protein. Statistical significance was determined using a two-sided unpaired *t*-test. The protein with a *p* value below 0.05 and a fold change above 1.5 or below 0.667 was considered differentially expressed.

Legends: Fig. 1 Comparative analysis of the spermathecal fluid proteome in

Aedes aegypti. **a** Volcano plot analysis of total proteins in three groups. Up-regulated, down-regulated, and non-significantly regulated proteins are marked with red, blue and grey spots, respectively. Candidates (LAP1 and DBF4) discussed in the text are highlighted (filled black triangles). LAP1: leucine aminopeptidase1 (AAEL006975-PB). DBF4: DBF4-type zinc finger (AAEL008779-PB). Fold change is the ratio between the mean quantitation values of protein. Statistical significance was determined using a two-sided unpaired *t*-test. **b** Venn diagram showing the uniquely and commonly regulated differentially expressed proteins (DEPs). The direction of arrows in the overlapping region represents up- and down-regulated proteins. **c** The Sankey dot plot of significantly up-regulated proteins in the G2/G1 group. Columns of Sankey represent proteins name (the left columns) and corresponding pathways (the right columns). Pathway names are shown below the plot. The sizes of dot plot indicate counts of proteins in the corresponding pathway and the colors represent the *p*-value. The DAVID web server was used for GO enrichment analysis and the *p*-value was determined by Fisher's exact test with Benjamini-Hochberg adjustment.

The experimental methods for Figure 2 are still unclear in the methods. We suggest creating a section for this figure in the methods section as has been done for Fig 5 and 6.

Answer:

Thank you for pointing this out. We added more details within the Methods as indicated below:

Mosquito fecundity and fertility: Mating assay as shown in Fig. 2a: Three days after microinjection, fifty females were mated with 50 males for 3 days. Females were then fed blood and transferred to separate oviposition tubes (10-mL centrifuge tube) for egg collection at 3 days PBM. The numbers of eggs were counted at 6 days PBM, and subsequently these eggs were used to calculate the hatchability. Female mosquitoes, dying or unwilling to have blood meal due to microinjection, were eliminated from the experiment.

Mating assay as shown in Fig. 2b-g: Mosquitoes were divided into three groups

($WT^{\text{♂}} \times WT^{\text{♀}}$, $LAPI^{-/-\text{♂}} \times LAPI^{-/-\text{♀}}$, and $LAPI^{-/+ \text{♂}} \times LAPI^{-/+ \text{♀}}$) and mated for 3 days. Fifty females were paired with 50 males per group. Females were then allowed to feed blood (the first blood meal) and moved to separate oviposition tubes for egg collection at 3 days PBM. After 3 days of oviposition, surviving females were transferred to a new container with water and 10% (w/v) sucrose for further blood meal⁶⁷. Eggs from the first blood meal were counted at 6 days PBM and subsequently these eggs were used to calculate hatchability. After three days of recovery, surviving females were given a blood feeding again (the second blood meal). They were then placed in oviposition tubes to lay eggs at 3 days PBM, and the numbers of eggs were counted at six days PBM. These eggs were used to calculate hatchability.

Mating assay as shown in Fig. 2h-m: Mosquitoes were separated into three groups ($WT^{\text{♂}} \times WT^{\text{♀}}$, $LAPI^{-/-\text{♂}} \times WT^{\text{♀}}$, and $WT^{\text{♂}} \times LAPI^{-/-\text{♀}}$) and mated for 3 days. Fifty females were paired with 50 males in each group. Egg deposition and hatchability were assessed after the first and second blood meals using the same mating assay as shown in Fig. 2b-g.

Also, for the analysis in Figure 2, it is indicated that a one-way ANOVA has been used. Based on what has been written here, we assume this is an ANOVA used to compare among the different crosses as. How were replicate effects taken into account? Which post-hoc pairwise comparison method was used (for example, Tukey's?) and how was this corrected for multiple comparisons?

Answer:

Thank you for pointing this out. First, we repeated each experiment twice independently in Fig. 2, which previously shows the results of only one independent experiment. A two-tailed Mann-Whitney test of variance was conducted to detect any significant variation between two replicates, and no significant differences were detected. Then results were combined for further analyses. Therefore, now the data were reproduced and combined from two independent experiments. Second, our data were not normally distributed, so a one-way ANOVA was not appropriate. Instead, we used the Kruskal-Wallis test in Fig.2 (a, b, c, d, e, h, i, j, and k). We added more details

within the legends as indicated below:

Fig. 2 *LAPI* mutation in males limits female fecundity and fertility. **a** Analysis of female fecundity and fertility after RNAi knockdown of ten candidate genes ($n > 37$). dsRNA was microinjected into the thorax of 1-day-old virgin female mosquitoes (50 females per group). After a 3-day recovery period, females were mated with 50 males for 3 days before the blood meal. The numbers of eggs were counted at 6 days PBM, and subsequently these eggs were used to calculate the hatchability. Microinjection resulted in a proportion of mosquitoes dying or being unwilling to have blood meal; they were removed from the experiment. Mosquitoes injected with *EGFP* dsRNA were used as controls. The q -value was calculated by comparing the experimental group with the *EGFP* dsRNA-treated group. **b-c** Comparison of the egg deposition among $WT^{\text{♂}} \times WT^{\text{♀}}$, $LAPI^{-/-\text{♂}} \times LAPI^{-/-\text{♀}}$, and $LAPI^{-/+ \text{♂}} \times LAPI^{-/+ \text{♀}}$ groups after the first (**b**; $n > 62$) and second (**c**; $n > 48$) blood meals (50 females were paired with 50 males per group). **d-g** Comparison of the hatchability among $WT^{\text{♂}} \times WT^{\text{♀}}$, $LAPI^{-/-\text{♂}} \times LAPI^{-/-\text{♀}}$, and $LAPI^{-/+ \text{♂}} \times LAPI^{-/+ \text{♀}}$ groups after the first (**d** and **f**; $n > 62$) and second (**e** and **g**; $n > 48$) blood meals (50 females were paired with 50 males per group). The percentages in **f** and **g** indicate the rate that mosquitoes produce hatchable eggs. **h-i** Comparison of the fecundity among $WT^{\text{♂}} \times WT^{\text{♀}}$, $LAPI^{-/-\text{♂}} \times WT^{\text{♀}}$, and $WT^{\text{♂}} \times LAPI^{-/-\text{♀}}$ groups after the first (**h**; $n > 62$) and second (**i**; $n > 54$) blood meals (50 females were paired with 50 males per group). **j-m** Comparison of the fertility among $WT^{\text{♂}} \times WT^{\text{♀}}$, $LAPI^{-/-\text{♂}} \times WT^{\text{♀}}$, and $WT^{\text{♂}} \times LAPI^{-/-\text{♀}}$ groups after the first (**j** and **l**; $n > 62$) and second (**k** and **m**; $n > 54$) blood meals (50 females were paired with 50 males per group). The percentages in **l** and **m** indicate the rate that mosquitoes produce hatchable eggs. The top of each column in **b**, **c**, **h** and **i** indicates the numbers of mosquitoes that laid eggs, while in **d**, **e**, **j** and **k** shows the numbers of hatched mosquitoes relative to total egg-laying mosquitoes. Females that died during egg laying were removed from the study. Each dot represents one mosquito. The experiments were independently performed twice with 50 females per experiment. A two-tailed Mann-Whitney test of variance was conducted to detect any significant variation between two replicates, and no significant

differences were detected. Then results were combined for further analyses. Egg deposition and hatchability were analyzed using the Kruskal-Wallis test followed by Dunn's post-hoc tests with Benjamini-Hochberg adjustment for multiple comparisons (**a**, **b**, **c**, **d**, **e**, **h**, **i**, **j**, and **k**). And Fisher's exact test was used to statistically analyze the ratio of females producing hatchable eggs relative to total mosquitoes (**f**, **g**, **l**, and **m**). Data in **a**, **b**, **c**, **d**, **e**, **h**, **i**, **j**, and **k** are shown as mean \pm SEM. * $q < 0.05$, ** $q < 0.01$, *** $q < 0.001$. ns: not significant.

We have similar questions about the analysis presented in Fig 5 and 6. How was replicate taken into account statistically and in Figure 5 how were interactions between cross type and male age or cross type and days between mating and blood feeding taken into account? Methods for post-hoc and corrections for multiple comparisons are required. Also, many of these outcome variables do not appear to be normally distributed which would make an ANOVA an inappropriate test. Please confirm data fit model assumptions.

Answer:

Thank you for pointing this out. A Kruskal-Wallis test of variance was conducted to detect any significant variation among three replicates, and no significant differences were detected. And, now the data were reproduced and combined from three independent experiments. Our data were not normally distributed, so an ANOVA was not appropriate. Instead, we used the multiple Mann-Whitney test in Fig.5 (**b** and **e**) and Kruskal-Wallis test in Fig.6 (**b** and **e**). We added more details within the legends as indicated below:

Fig. 5 Mating with *LAPI*^{-/-} males suppresses female fertility. **a** Schematic diagram of the stored sperm fertility assay. After mating with 1-day-old WT or *LAPI*^{-/-} males for 3 days, WT females were blood fed at indicated days (0, 4, 8, 12, and 16 days) post mating. And egg hatchability was measured. **b** The fertility of WT females following blood feeding at indicated days after mating with 1-day-old WT or *LAPI*^{-/-} males (30 females were paired with 30 males per group) (n>55). The *q*-value was

calculated by comparing the experimental group with the 0 days post mating group. **c** Curve fitting of the same data from **b**. The dashed line shows the point, at which mosquitoes were blood fed after mating with *LAPI*^{-/-} males resulted in a 10% of female fertility. The red curve was served as the fitted curve. **d** Schematic diagram of the sperm fertility assay. WT females were mated with indicated day-old (1-, 3-, 5-, 7-, 9-, 11-day-old) WT or *LAPI*^{-/-} males. And blood meal was given after mating for 3 days. Egg deposition and hatchability were measured later. **e** The fertility of WT females mated with indicated old-WT or *LAPI*^{-/-} males (30 females were paired with 30 males per group) (n>57). The *q*-value was calculated by comparing the experimental group with the 1-day-old group. **f** Curve fitting of the same data from **e**. The dashed line shows the point at which mating with indicated old-*LAPI*^{-/-} males resulted in a 10% reduction in female fertility. The red curve was represented the fitted curve. Females that died during laying eggs were removed from the study. Three days after oviposition, the numbers of eggs were counted and subsequently separated into intact (unhatched) or broken (hatched). The top of each column in **b** and **e** shows the number of hatched mosquitoes relative to total mosquitoes. Each dot represents one mosquito. The experiments were independently performed three times with 30 females per experiment. A Kruskal-Wallis test of variance was conducted to detect any significant variation among three replicates, and no significant differences were detected. Then results were combined for further analyses. Statistical significance was determined using the multiple Mann-Whitney test with Benjamini-Hochberg adjustment (**b** and **e**). Data are shown as mean \pm SEM. * *q* < 0.05, ** *q* < 0.01, *** *q* < 0.001.

Fig. 6 *LAPI*^{-/-} males compete with WT males to limit female fertility. **a** Scheme of the mating competition assay of 1-day-old male mosquitoes (30 females were paired with 30 males per group). The ratios of *LAPI*^{-/-}♂ to WT ♂ were 0:1 (0 *LAPI*^{-/-}♂ and 30 WT ♂), 1:1 (15 *LAPI*^{-/-}♂ and 15 WT ♂), 2:1 (20 *LAPI*^{-/-}♂ and 10 WT ♂), 5:1 (25 *LAPI*^{-/-}♂ and 5 WT ♂), 9:1 (27 *LAPI*^{-/-}♂ and 3 WT ♂), 14:1 (28 *LAPI*^{-/-}♂ and 2 WT ♂), and 29:1 (29 *LAPI*^{-/-}♂ and 1 WT ♂) for a total of 30 males in each group. **b** The fertility of WT females mated with 1-day-old *LAPI*^{-/-} and WT males at different ratios

was evaluated ($n > 56$). 1-day-old *LAPI*^{-/-} and WT males were introduced simultaneously at different ratios (30 males in total) to mate with 30 WT females, and females were given blood meal after 3 days. Egg hatchability was then measured. **c** Curve fitting of the same data from **b**. The dashed line shows the point at which mating with 1-day-old males of indicated *LAPI*^{-/-} to WT ratios resulted in a 50% reduction in female fertility. The fitted curve was marked with red. **d** Scheme of mating competition assay of 6-day-old male mosquitoes (30 females were paired with 30 males per group). **e** The fertility of WT females mated with *LAPI*^{-/-} and WT males at different ratios. 6-day-old *LAPI*^{-/-} and 1-day-old WT males were introduced simultaneously at different ratios (30 males in total) to mate with 30 WT females ($n > 58$). Blood meal was given after 3 days. And egg hatchability was measured. **f** Curve fitting of the same data from **e**. The dashed line shows the point at which mating with 6-day-old males of indicated *LAPI*^{-/-} to WT ratios resulted in a 50% reduction in female fertility. The fitted curve was marked with red. Females that died during laying eggs were removed from the study. The top of each column in **b** and **e** shows the number of hatched mosquitoes relative to total mosquitoes. Each dot represents one mosquito. The experiments were independently performed three times with 30 females per experiment. A Kruskal-Wallis test of variance was conducted to detect any significant variation among three replicates, and no significant differences were detected. Then results were combined for further analyses. Statistical significance of hatchability was determined using the Kruskal-Wallis test followed by Dunn's post-hoc tests with Benjamini-Hochberg adjustment for multiple comparisons (**b** and **e**). The *q*-value was calculated by comparing the experimental group with the 0:1 group. Data are shown as mean \pm SEM. * $q < 0.05$, ** $q < 0.01$, *** $q < 0.001$.

The information in the statistical analysis section (MS Line 702-705) is so vague that it is not informative. We suggest describing each statistical approach with the corresponding section of experimental methods.

Answer:

We added more details within the methods:

Statistical analysis: Proteomic data were performed using <http://www.bioinformatics.com.cn>, which is used for analysis and visualization (Fig. 1a, 1b, 1c, Supplementary Fig. 1b, and 1c). Volcano plot analysis in Fig. 1a, GO enrichment analysis in Supplementary Fig. 1b, KEGG functional classification in Supplementary Fig. 1c, and the survival rate in Supplementary Fig. 5c were performed by enhancedvolcano R package (1.13.2), Goplot R package (1.0.2), circlize R package (0.4.15), and survminer R package (0.4.9) in the R environment, respectively. Venn diagram in Fig. 1b and Sankey dot plot in Fig. 1c were performed using the matplotlib python package (3.7.0). Statistical analyses of the remaining data were done by GraphPad Prism statistical software. Data are shown as mean \pm SEM. Statistical significance is provided in the figure legends. Statistical significance of egg deposition and hatchability in Fig.2 and Fig.6 was determined using the Kruskal-Wallis test followed by Dunn's post-hoc tests with Benjamini-Hochberg adjustment for multiple comparisons. Fisher's exact test was used to statistically analyze the ratio of females producing hatchable eggs to total mosquitoes. Statistical significance of the mRNA abundance of *LAPI* in Fig. 3a were determined using the one-way ANOVA test followed by Tukey's post hoc tests with Benjamini-Hochberg adjustment for multiple comparisons. Statistical significance of the expression of *LAPI* in Fig. 3d and hatchability in Fig.5 was determined using the multiple Mann-Whitney test with Benjamini-Hochberg adjustment. The *p*-value of the length of the ovarian follicles in Fig. 3c was determined by two-tailed Mann-Whitney test. For curve fitting, all error bars represent \pm SEM (Fig. 5c, 5f, 6c and 6f), which is 95% CI to indicate that the fitted curves are in the range of all error bars.

There are several places throughout the MS where the term "significant difference" is used without statistical support. Instead, please use a different term here such as "large" or "notable". Significance should only be used in tandem with statistical analysis.

Answer:

These are excellent suggestions and we have corrected as suggested throughout

the manuscript.

We changed “The density of LAP1 immunofluorescence, however, was significantly lower in *LAPI*^{-/+} male mosquitoes than WT” to “The density of LAP1 immunofluorescence, however, was notably lower in *LAPI*^{-/+} male mosquitoes than WT”;

We changed “Importantly, the addition of 29:1 resulted in a significant additional reduction” to “Importantly, the addition of 29:1 resulted in a notable reduction”.

Thank you for including clarification of the sample sizes. To keep this clear, please give the planned sample size in the methods for each experiment (30 or 50) and the realized sample size in the figure legend (as you have done) with your clarification that many females (⅔) are lost in all assays.

Answer:

Thank you for pointing this out. Previously, the data were the results of one independent experiment. Now, a two-tailed Mann-Whitney test of variance was conducted to detect any significant variation between two replicates and a Kruskal-Wallis test was used among three replicates, and no significant differences were detected. Then results were pooled for further analyses. Therefore, the data are the replicated and combined results of two or three independent experiments. We added more details within the legends (Fig. 2a-m; Fig.3a, 3c and 3d; Fig. 5b and 5e; Fig. 6a, 6b, 6d, and 6e) as indicated below:

Fig. 2 *LAPI* mutation in males limits female fecundity and fertility. **a** Analysis of female fecundity and fertility after RNAi knockdown of ten candidate genes (n>37). dsRNA was microinjected into the thorax of 1-day-old virgin female mosquitoes (50 females per group). After a 3-day recovery period, females were mated with 50 males for 3 days before the blood meal. The numbers of eggs were counted at 6 days PBM, and subsequently these eggs were used to calculate the hatchability. Microinjection resulted in a proportion of mosquitoes dying or being unwilling to have blood meal; they were removed from the experiment. Mosquitoes injected with *EGFP* dsRNA were

used as controls. The q -value was calculated by comparing the experimental group with the *EGFP* dsRNA-treated group. **b-c** Comparison of the egg deposition among $WT^{\♂} \times WT^{\♀}$, $LAPI^{-/-\♂} \times LAPI^{-/-\♀}$, and $LAPI^{-/+ \♂} \times LAPI^{-/+ \♀}$ groups after the first (**b**; $n > 62$) and second (**c**; $n > 48$) blood meals (50 females were paired with 50 males per group). **d-g** Comparison of the hatchability among $WT^{\♂} \times WT^{\♀}$, $LAPI^{-/-\♂} \times LAPI^{-/-\♀}$, and $LAPI^{-/+ \♂} \times LAPI^{-/+ \♀}$ groups after the first (**d** and **f**; $n > 62$) and second (**e** and **g**; $n > 48$) blood meals (50 females were paired with 50 males per group). The percentages in **f** and **g** indicate the rate that mosquitoes produce hatchable eggs. **h-i** Comparison of the fecundity among $WT^{\♂} \times WT^{\♀}$, $LAPI^{-/-\♂} \times WT^{\♀}$, and $WT^{\♂} \times LAPI^{-/-\♀}$ groups after the first (**h**; $n > 62$) and second (**i**; $n > 54$) blood meals (50 females were paired with 50 males per group). **j-m** Comparison of the fertility among $WT^{\♂} \times WT^{\♀}$, $LAPI^{-/-\♂} \times WT^{\♀}$, and $WT^{\♂} \times LAPI^{-/-\♀}$ groups after the first (**j** and **l**; $n > 62$) and second (**k** and **m**; $n > 54$) blood meals (50 females were paired with 50 males per group). The percentages in **l** and **m** indicate the rate that mosquitoes produce hatchable eggs. The top of each column in **b**, **c**, **h** and **i** indicates the numbers of mosquitoes that laid eggs, while in **d**, **e**, **j** and **k** shows the numbers of hatched mosquitoes relative to total egg-laying mosquitoes. Females that died during egg laying were removed from the study. Each dot represents one mosquito. The experiments were independently performed twice with 50 females per experiment. A two-tailed Mann-Whitney test of variance was conducted to detect any significant variation between two replicates, and no significant differences were detected. Then results were combined for further analyses. Egg deposition and hatchability were analyzed using the Kruskal-Wallis test followed by Dunn's post-hoc tests with Benjamini-Hochberg adjustment for multiple comparisons (**a**, **b**, **c**, **d**, **e**, **h**, **i**, **j**, and **k**). And Fisher's exact test was used to statistically analyze the ratio of females producing hatchable eggs relative to total mosquitoes (**f**, **g**, **l**, and **m**). Data in **a**, **b**, **c**, **d**, **e**, **h**, **i**, **j**, and **k** are shown as mean \pm SEM. * $q < 0.05$, ** $q < 0.01$, *** $q < 0.001$. ns: not significant.

Fig. 3 The localization of LAPI in *Ae. aegypti*. **a** mRNA abundance of *LAPI* in different mosquito tissue at the G2 ($n=3$). The experiment was replicated three times,

and tissues collected from 20 individual females and males at 72 h PE were served as a single replicate per group. Data were normalized to the expression level of ovary♀. Statistical significance was determined using the one-way ANOVA test followed by Tukey's post hoc tests with Benjamini-Hochberg adjustment for multiple comparisons. *** $q < 0.001$. **b** Images of sperm in WT, $LAPI^{-/+}$, and $LAPI^{-/-}$ male mosquitoes. Sperm was dissected, and LAP1 protein was detected by immunofluorescence assay using mouse anti-LAP1 antibody (green). Nucleus was stained by Hoechst 33258 (blue). Images were visualized under a confocal microscope. Scale bar: 20 μm . **c** The ovarian phenotypes (left panel) and follicle lengths (right panel) of WT and $LAPI^{-/-}$ female mosquitoes were detected at 48 h PBM ($n > 26$). Results are combined from three batches of mosquitoes, and each dot represents the length of a single follicle. Images were captured from CellSens software (version 1.6) using an Olympus SZX16 stereoscopic microscope at 5 \times magnification (scale bar: 500 μm). The follicle lengths were measured in CellSens software (version 1.6). The p -value was determined by two-tailed Mann-Whitney test. **d** Temporal expression of $LAPI$ during different stages of embryonic development in $LAPI^{-/-}\text{♂} \times \text{WT}\text{♀}$ and $\text{WT}\text{♂} \times \text{WT}\text{♀}$ groups ($n=3$). The experiment was repeated three times, and total RNA was extracted from eggs collected from the pairs $LAPI^{-/-}\text{♂} \times \text{WT}\text{♀}$ and $\text{WT}\text{♂} \times \text{WT}\text{♀}$ at different periods (0-1, 1-2, 2-4, 4-6, 6-8, 8-10, 10-12, and 20-22 h) after the first and second blood meals as individual replicates for each group. RNA level of $LAPI$ was measured using qPCR. Data were normalized to the expression level of $\text{WT}\text{♂} \times \text{WT}\text{♀}$ at 0-1 h post first blood meal. Statistical significance was determined using the multiple Mann-Whitney test with Benjamini-Hochberg adjustment. ns: not significant. Data in **a**, **c** and **d** are represented as mean \pm SEM. The experiments were repeated three times with similar results.

Fig. 5 Mating with $LAPI^{-/-}$ males suppresses female fertility. **a** Schematic diagram of the stored sperm fertility assay. After mating with 1-day-old WT or $LAPI^{-/-}$ males for 3 days, WT females were blood fed at indicated days (0, 4, 8, 12, and 16 days) post mating. And egg hatchability was measured. **b** The fertility of WT females following blood feeding at indicated days after mating with 1-day-old WT or $LAPI^{-/-}$

males (30 females were paired with 30 males per group) ($n > 55$). The q -value was calculated by comparing the experimental group with the 0 days post mating group. **c** Curve fitting of the same data from **b**. The dashed line shows the point, at which mosquitoes were blood fed after mating with $LAPI^{-/-}$ males resulted in a 10% of female fertility. The red curve was served as the fitted curve. **d** Schematic diagram of the sperm fertility assay. WT females were mated with indicated day-old (1-, 3-, 5-, 7-, 9-, 11-day-old) WT or $LAPI^{-/-}$ males. And blood meal was given after mating for 3 days. Egg deposition and hatchability were measured later. **e** The fertility of WT females mated with indicated old-WT or $LAPI^{-/-}$ males (30 females were paired with 30 males per group) ($n > 57$). The q -value was calculated by comparing the experimental group with the 1-day-old group. **f** Curve fitting of the same data from **e**. The dashed line shows the point at which mating with indicated old- $LAPI^{-/-}$ males resulted in a 10% reduction in female fertility. The red curve was represented the fitted curve. Females that died during laying eggs were removed from the study. Three days after oviposition, the numbers of eggs were counted and subsequently separated into intact (unhatched) or broken (hatched). The top of each column in **b** and **e** shows the number of hatched mosquitoes relative to total mosquitoes. Each dot represents one mosquito. The experiments were independently performed three times with 30 females per experiment. A Kruskal-Wallis test of variance was conducted to detect any significant variation among three replicates, and no significant differences were detected. Then results were combined for further analyses. Statistical significance was determined using the multiple Mann-Whitney test with Benjamini-Hochberg adjustment (**b** and **e**). Data are shown as mean \pm SEM. * $q < 0.05$, ** $q < 0.01$, *** $q < 0.001$.

Fig. 6 $LAPI^{-/-}$ males compete with WT males to limit female fertility. **a** Scheme of the mating competition assay of 1-day-old male mosquitoes (30 females were paired with 30 males per group). The ratios of $LAPI^{-/-}$ ♂ to WT ♂ were 0:1 (0 $LAPI^{-/-}$ ♂ and 30 WT ♂), 1:1 (15 $LAPI^{-/-}$ ♂ and 15 WT ♂), 2:1 (20 $LAPI^{-/-}$ ♂ and 10 WT ♂), 5:1 (25 $LAPI^{-/-}$ ♂ and 5 WT ♂), 9:1 (27 $LAPI^{-/-}$ ♂ and 3 WT ♂), 14:1 (28 $LAPI^{-/-}$ ♂ and 2 WT ♂), and 29:1 (29 $LAPI^{-/-}$ ♂ and 1 WT ♂) for a total of 30 males in each group. **b** The

fertility of WT females mated with 1-day-old *LAPI*^{-/-} and WT males at different ratios was evaluated (n>56). 1-day-old *LAPI*^{-/-} and WT males were introduced simultaneously at different ratios (30 males in total) to mate with 30 WT females, and females were given blood meal after 3 days. Egg hatchability was then measured. **c** Curve fitting of the same data from **b**. The dashed line shows the point at which mating with 1-day-old males of indicated *LAPI*^{-/-} to WT ratios resulted in a 50% reduction in female fertility. The fitted curve was marked with red. **d** Scheme of mating competition assay of 6-day-old male mosquitoes (30 females were paired with 30 males per group). **e** The fertility of WT females mated with *LAPI*^{-/-} and WT males at different ratios. 6-day-old *LAPI*^{-/-} and 1-day-old WT males were introduced simultaneously at different ratios (30 males in total) to mate with 30 WT females (n>58). Blood meal was given after 3 days. And egg hatchability was measured. **f** Curve fitting of the same data from **e**. The dashed line shows the point at which mating with 6-day-old males of indicated *LAPI*^{-/-} to WT ratios resulted in a 50% reduction in female fertility. The fitted curve was marked with red. Females that died during laying eggs were removed from the study. The top of each column in **b** and **e** shows the number of hatched mosquitoes relative to total mosquitoes. Each dot represents one mosquito. The experiments were independently performed three times with 30 females per experiment. A Kruskal-Wallis test of variance was conducted to detect any significant variation among three replicates, and no significant differences were detected. Then results were combined for further analyses. Statistical significance of hatchability was determined using the Kruskal-Wallis test followed by Dunn's post-hoc tests with Benjamini-Hochberg adjustment for multiple comparisons (**b** and **e**). The *q*-value was calculated by comparing the experimental group with the 0:1 group. Data are shown as mean ± SEM. * *q* < 0.05, ** *q* < 0.01, *** *q* < 0.001.

In the 'safety and fitness assessments of LAP1^{-/-} males' experiments it isn't clear how the ratios of males of different genotypes was achieved; a ratio of 3:1 WT to LAP1^{-/-} males (MS line 317-319) isn't possible with 30 males in total.

Answer:

Thanks for your suggestions. This is a mistake. We added more details within the legends (Fig.6a) as indicated below:

Fig. 6 *LAPI*^{-/-} males compete with WT males to limit female fertility. **a** Scheme of the mating competition assay of 1-day-old male mosquitoes (30 females were paired with 30 males per group). The ratios of *LAPI*^{-/-}♂ to WT ♂ were 0:1 (0 *LAPI*^{-/-}♂ and 30 WT ♂), 1:1 (15 *LAPI*^{-/-}♂ and 15 WT ♂), 2:1 (20 *LAPI*^{-/-}♂ and 10 WT ♂), 5:1 (25 *LAPI*^{-/-}♂ and 5 WT ♂), 9:1 (27 *LAPI*^{-/-}♂ and 3 WT ♂), 14:1 (28 *LAPI*^{-/-}♂ and 2 WT ♂), and 29:1 (29 *LAPI*^{-/-}♂ and 1 WT ♂) for a total of 30 males in each group. **b** The fertility of WT females mated with 1-day-old *LAPI*^{-/-} and WT males at different ratios was evaluated (n>56). 1-day-old *LAPI*^{-/-} and WT males were introduced simultaneously at different ratios (30 males in total) to mate with 30 WT females, and females were given blood meal after 3 days. Egg hatchability was then measured. **c** Curve fitting of the same data from **b**. The dashed line shows the point at which mating with 1-day-old males of indicated *LAPI*^{-/-} to WT ratios resulted in a 50% reduction in female fertility. The fitted curve was marked with red. **d** Scheme of mating competition assay of 6-day-old male mosquitoes (30 females were paired with 30 males per group). **e** The fertility of WT females mated with *LAPI*^{-/-} and WT males at different ratios. 6-day-old *LAPI*^{-/-} and 1-day-old WT males were introduced simultaneously at different ratios (30 males in total) to mate with 30 WT females (n>58). Blood meal was given after 3 days. And egg hatchability was measured. **f** Curve fitting of the same data from **e**. The dashed line shows the point at which mating with 6-day-old males of indicated *LAPI*^{-/-} to WT ratios resulted in a 50% reduction in female fertility. The fitted curve was marked with red. Females that died during laying eggs were removed from the study. The top of each column in **b** and **e** shows the number of hatched mosquitoes relative to total mosquitoes. Each dot represents one mosquito. The experiments were independently performed three times with 30 females per experiment. A Kruskal-Wallis test of variance was conducted to detect any significant variation among three replicates, and no significant differences were detected. Then results were combined for further analyses. Statistical significance of hatchability was determined using the

Kruskal-Wallis test followed by Dunn's post-hoc tests with Benjamini-Hochberg adjustment for multiple comparisons (**b** and **e**). The q -value was calculated by comparing the experimental group with the 0:1 group. Data are shown as mean \pm SEM.
* $q < 0.05$, ** $q < 0.01$, *** $q < 0.001$.

Reviewers' Comments:

Reviewer #2:

Remarks to the Author:

The authors have greatly improved the clarity of the manuscript. I have some minor comments for them to address regarding the statistical analysis section

Minor comments:

Line 429-453: Three points seem to be intermingled here and it making it challenging to keep track of the rationale for each. One is the LAP1 +/- lines could be more easily maintained as homozygous lines which is clear. The second point about the residual fertility possible being an issue is clear as well and the authors discuss future improvements. The third point about the residual fertility negating the need for sustained release for population suppression is not clearly presented. If this is kept in the discussion it needs to be laid out more convincingly. Whichever the authors decide, these three points need to be presented separately.

Line 701-704: Some details of the survival and GO enrichment analysis are not clear. The documentation for Goplot indicates that it is for visualising data, but doesn't explain the analysis used. "More precisely, the package will help combine and integrate expression data with the results of a functional analysis. The package cannot be used to perform any of these analyses. It is for visualization purpose only." The significance of enrichment analysis is not a key output of the study and would not change the overall conclusions, but it would be good to clarify how the Goplot package is being used here. Similarly, the survminer R package looks to have include multiple types of survival analysis (Cox regression, Kaplan Meier). I think for looking at the legend of Fig S5 and MS Line 515, that the authors have plotted Kaplan Meier curves and used a Mantel-Cox test to compare groups. This needs to be clarified in the statistical analysis section.

Suggestions:

Line 76- remove "and"

Line 305-309: It wasn't clear to me in what context blood feeding time of females or time when females mated with males would be controlled. Is this referring to within a mass rearing operation or somehow during a field release?

Line 363: Remove "Consistently,"

Line 340: replace noticeable with significant- "there was no significant effect" and give P-value from the Mantel-Cox test here or on Figure S5.

Lines 380-390: This passage was a little difficult to follow. Consider editing for clarity. For example: "Concordant with previous studies, we also found that LAPs were significantly upregulated after insemination and enriched on pathways.....suggesting that LAPs serve..."

Throughout discussion cite figures where appropriate result is discussed.

Line 416- 419: "Nevertheless, traditional SIT relies on radiation or .." I would say that measured effects of appropriate levels of radiation are variable (see Bellini et al 2021, J Med Ent, 58(2):807-813 for example) and no worse than those reported for genetic SIT approaches (Harris et al. 2012 Nature Biotechnology 30 (9): 828-830, Carvalho et al. 2015 PLoS NTD 9(7):e0003864. I suggest editing this sentence to not specifically point to chemical SIT.

Line 476-479: Small edit need to clarify which groups are mated and virgin as in Lines 116-119.

Line 515: Need some clarification on this analysis Kaplan Meier survival is plotted but the Mantel-Cox is used for comparing groups, correct?

Line 615: Thank you for clarifying ratios!

REVIEWER COMMENTS

Line 429-453: Three points seem to be intermingled here and it making it challenging to keep track of the rationale for each. One is the *LAP1* $-/-$ lines could be more easily maintained as homozygous lines which is clear. The second point about the residual fertility possible being an issue is clear as well and the authors discuss future improvements. The third point about the residual fertility negating the need for sustained release for population suppression is not clearly presented. If this is kept in the discussion it needs to be laid out more convincingly. Whichever the authors decide, these three points need to be presented separately.

Answer:

Thank you for pointing this out. We changed in the discussion. We deleted the third point.

“Moreover, in contrast with the previous SIT in which sterile males could not be maintained as homozygous lines, *LAP1* mutants are able to generate hatchable offspring at an average of 9.5% after the first blood meal. Ideally, genetic changes arising from sterility-inducing genes should be transmitted through insect populations without relying on a sustained mass release of sterile insects, which is the concept behind the development of the CRISPR/Cas9 gene editing system⁵⁹. Therefore, mating with *LAP1*^{-/-} males is effective in spreading genetic changes to mosquito populations through a low level of fertilization and consequent low number of hatchable offspring. Importantly, the fertilizing ability of sperm in *LAP1*^{-/-} males decreased over time (Fig. 5e). Thus, we were able to control populations by mating with *LAP1*^{-/-} males at an appropriate age.

Nonetheless, high residual fertility might affect the political and ethical acceptability of this technology, particularly in areas where mosquito-borne diseases are endemic, and it is recommended that it be kept below 1%⁶². Thus, one of the next steps in the use of *LAP1*^{-/-} males for SIT is to overcome the challenge of high residual fertility. Our spermathecal fluid proteome data showed that in addition to *LAP1*, which was significantly higher in the G2/G1 group, three LAPs (AAEL000424-PA, AAEL001649-PA, and AAEL002978-PA) were observed in this group (Supplementary Table 1). We could study the effect of these LAPs on male sperm and reduce mosquito fertility by double or multiple knockdowns. Most important, transferring laboratory results to practical applications requires further research and work, including

investigation of release ratios, environmental conditions, and regulatory strategies, to ensure that the function of population suppression in the field is as effective and safe as that in the laboratory.

Overall, it is conceivable that mosquito populations and vector borne diseases could be reduced by releasing *LAPI*^{-/-} males at an indicated age within a certain range that is neither ecologically damaging nor harmful to human health. Moreover, the amino acid sequence of *Ae. aegypti* LAPI is highly conserved in *Aedes albopictus* and *Culex quinquefasciatus*, with 97% and 91% identities, respectively, indicating that strategies for deleting *LAPI* can be applied to other mosquito species to improve its efficacy in controlling a wide range of mosquito populations. Thus, our work provides a target gene for the gene drive system, further amplifying the function of LAPI in reducing mosquito populations.”

Line 701-704: Some details of the survival and GO enrichment analysis are not clear. The documentation for Goplot indicates that it is for visualising data, but doesn't explain the analysis used. “More precisely, the package will help combine and integrate expression data with the results of a functional analysis. The package cannot be used to perform any of these analyses. It is for visualization purpose only.” The significance of enrichment analysis is not a key output of the study and would not change the overall conclusions, but it would be good to clarify how the Goplot package is being used here. Similarly, the survminer R package looks to have include multiple types of survival analysis (Cox regression, Kaplan Meier). I think for looking at the legend of Fig S5 and MS Line 515, that the authors have plotted Kaplan Meier curves and used a Mantel-Cox test to compare groups. This needs to be clarified in the statistical analysis section.

Answer:

Thank you for pointing this out. We have previously explained the analysis used for the GO and KEGG enrichment analysis in the legends of Fig. 1 and Supplementary Fig. 1, as shown below:

“The DAVID web server was used for GO enrichment analysis and the *p*-value was determined by Fisher's exact test with Benjamini-Hochberg adjustment.” “The KOBAS web server was used for KEGG enrichment analysis and the *p*-value was determined by Fisher's exact test with Benjamini-Hochberg adjustment.”

Now, we added more details within the methods (Statistical analysis), as indicated below:

The Kaplan-Meier curves in Supplementary Fig. 5c were plotted using survival (3.5-7) and survminer R package (0.4.9), and the *p*-value was calculated using log-rank test (Mantel-Cox test). The DAVID web server (<https://david.ncifcrf.gov/>) was used for GO enrichment analysis and the *p*-value was determined by Fisher's exact test with Benjamini-Hochberg adjustment. KEGG enrichment analysis was performed using KOBAS web server (<http://bioinfo.org/kobas/>) and the *p*-value was tested by Fisher's exact test with Benjamini-Hochberg adjustment.

Line 76- remove “and”

Answer:

Thank you for pointing this out. We corrected as suggested.

Line 305-309: It wasn't clear to me in what context blood feeding time of females or time when females mated with males would be controlled. Is this referring to within a mass rearing operation or somehow during a field release?

Answer:

Thanks for pointing this out. In this paper, we have demonstrated that blood feeding time of females or time when females mated with males could be controlled under a mass rearing operation under laboratory conditions, while we could not control it in the field release.

Line 363: Remove “Consistently,”

Answer:

Thank you for pointing this out. We corrected as suggested.

Line 340: replace noticeable with significant- “there was no significant effect” and give P-value from the Mantel-Cox test here or on Figure S5.

Answer:

We corrected as suggested.

Lines 380-390: This passage was a little difficult to follow. Consider editing for clarity. For example: “Concordant with previous studies, we also found that LAPs were significantly upregulated after insemination and enriched on pathways.....suggesting that LAPs serve...”

Answer:

Thank you for pointing this out. We corrected as suggested in the discussion, as shown below:

“Concordant with previous studies, we also found that LAPs were significantly upregulated after insemination and enriched in GO terms—manganese ion binding, metalloexopeptidase activity, cytoplasm, and aminopeptidase activity—suggesting that LAPs serve a critical role in peptide turnover (Fig. 1c). In *Drosophila*, the LAP family was expanded and constituted the primary protein components of sperm¹⁸. Although LAP may not exert peptide turnover role in spermatogenesis due to loss of enzymatic activity during evolution, LAP is essential for the normal development of sperm²¹. In this study, we demonstrated that *LAPI* deficiency results in lower egg deposition, and this might occur due to poor blood digestion in the midgut (Fig. 2b and Fig.3c)⁵⁰.”

Throughout discussion cite figures where appropriate result is discussed.

Answer:

We made a modification in Discussion as suggested:

“Our proteomics-based analysis of mosquito spermathecae proteins has implied that the presence and content of sperm influence the expression of many proteins in the spermathecae (Fig. 1a).”

“Furthermore, functional annotation of total protein cohorts indicated enrichment

of categories related to cytoskeleton and microtubule synthesis in GO analysis, as well as identified phagosome, metabolic, and glycolysis/gluconeogenesis pathways by KEGG analysis (Supplementary Fig. 1b and 1c).”

“Concordant with previous studies, we also found that LAPs were significantly upregulated after insemination and enriched on pathways—manganese ion binding, metalloexopeptidase activity, cytoplasm, and aminopeptidase activity—suggesting that LAPs serve a critical role in peptide turnover (Fig. 1c).”

“In this study, we demonstrated that *LAPI* deficiency results in lower egg deposition, and this might occur due to poor blood digestion in the midgut (Fig. 2b and Fig.3c)⁵⁰.”

“When females mated with them, hatchability was markedly reduced after both first and second blood meals (Fig. 2d and 2e).”

“Our studies showed that *LAPI* localized to the sperm mitochondria, and its knockout resulted in defective mitochondrial derivatives (Fig. 3b).”

“Unlike the findings in other species, we discovered that the 9 + 9 + 1 microtubule structure of sperm dissociated after *LAPI* knockout in mosquitoes (Fig. 4b).”

“In our study, we used the CRISPR/Cas9 system to generate low-fertility *LAPI*^{-/-} males that show no differences from WT males in terms of physiological parameters under laboratory conditions, including longevity and virus transmission capacity (Supplementary Fig. 5a and 5b),”

“Moreover, in contrast with the previous SIT in which sterile males could not be maintained as homozygous lines, *LAPI* mutants are able to generate hatchable offspring at an average of 9.5% after the first blood meal (Fig. 2d).”

“Importantly, the fertilizing ability of sperm in *LAPI*^{-/-} males decreased over time (Fig. 5e).”

Line 416- 419: “Nevertheless, traditional SIT relies on radiation or ..” I would say that measured effects of appropriate levels of radiation are variable (see Bellini et al 2021, J Med Ent, 58(2):807-813 for example) and no worse than those reported for genetic SIT approaches (Harris et al. 2012 Nature

Biotechnology 30 (9): 828-830, Carvalho et al. 2015 PLoS NTD 9(7):e0003864.

I suggest editing this sentence to not specifically point to chemical SIT.

Answer:

Thank you for pointing this out. We corrected as suggested in the discussion, as shown below:

“Nevertheless, traditional SIT produces sterile males that possess low mating competitiveness due to mutagenesis of many genes⁵⁷.”

Line 476-479: Small edit need to clarify which groups are mated and virgin as in Lines 116-119.

Answer:

Thank you for pointing this out. We corrected as suggested in the methods (SWATH-MS), as indicated below:

“Spermathecae from 2000 individual female mosquitoes at the virgin (unmated) at 72 h PE, mated at 72 h PE and mated at 5 days PBM stages were pooled for the construction of a spectral library.”

Line 515: Need some clarification on this analysis Kaplan Meier survival is plotted but the Mantel-Cox is used for comparing groups, correct?

Answer:

Thank you for pointing this out. Yes, this is correct. We added more details in the methods (Mosquito lifespan) such as below:

“Kaplan Meier (KM) survival curves were plotted using <http://www.bioinformatics.com.cn/en>, an online platform for data analysis and visualization. The *p*-value was calculated using log-rank test (Mantel-Cox test).”

Line 615: Thank you for clarifying ratios!

Answer:

Thank you.